# Remove that Square Root: A New Efficient Scale-Invariant Version of AdaGrad

**Sayantan Choudhury**[*]
MBZUAI & Johns Hopkins University

**Nazarii Tupitsa**
MBZUAI & Innopolis University

**Nicolas Loizou**
Johns Hopkins University

**Samuel Horváth**
MBZUAI

**Martin Takáč**
MBZUAI

**Eduard Gorbunov**
MBZUAI

## Abstract

Adaptive methods are extremely popular in machine learning as they make learning rate tuning less expensive. This paper introduces a novel optimization algorithm named KATE, which presents a scale-invariant adaptation of the well-known AdaGrad algorithm. We prove the scale-invariance of KATE for the case of Generalized Linear Models. Moreover, for general smooth non-convex problems, we establish a convergence rate of $\mathcal{O}(\log T/\sqrt{T})$ for KATE, matching the best-known ones for AdaGrad and Adam. We also compare KATE to other state-of-the-art adaptive algorithms Adam and AdaGrad in numerical experiments with different problems, including complex machine learning tasks like image classification and text classification on real data. The results indicate that KATE consistently outperforms AdaGrad and matches/surpasses the performance of Adam in all considered scenarios.

## 1 Introduction

In this work, we consider the following unconstrained optimization problem:

$$\min_{w \in \mathbb{R}^d} f(w), \tag{1}$$

where $f : \mathbb{R}^d \to \mathbb{R}$ is a $L$-smooth and generally non-convex function. In particular, we are interested in the situations when the objective has either expectation $f(w) = \mathbb{E}_{\xi \sim \mathcal{D}}[f_\xi(w)]$ or finite-sum $f(w) = \frac{1}{n} \sum_{i=1}^{n} f_i(w)$ form. Such minimization problems are crucial in machine learning, where $w$ corresponds to the model parameters. Solving these problems with stochastic gradient-based optimizers has gained much interest owing to their wider applicability and low computational cost. Stochastic Gradient Descent (SGD) (Robbins and Monro, 1951) and similar algorithms require the knowledge of parameters like $L$ for convergence and are very sensitive to the choice of the stepsize in general. Therefore, SGD requires hyperparameter tuning, which can be computationally expensive. To address these issues, it is common practice to use adaptive variants of stochastic gradient-based methods that can converge without knowing the function's structure.

There exist many adaptive algorithms such as AdaGrad (Duchi et al., 2011), Adam (Kingma and Ba, 2014), AMSGrad (Reddi et al., 2019), D-Adaptation (Defazio and Mishchenko, 2023), Prodigy (Mishchenko and Defazio, 2023), AI-SARAH (Shi et al., 2023) and their variants. These adaptive techniques are capable of updating their step sizes on the fly. For instance, the AdaGrad method determines its step sizes using a cumulative sum of the coordinate-wise squared (stochastic) gradient

---

[*]Part of this work was done when S. Choudhury was an intern at MBZUAI, UAE.

38th Conference on Neural Information Processing Systems (NeurIPS 2024).

of all the previous iterates:

$$\text{AdaGrad: } w_{t+1} = w_t - \frac{\beta g_t}{\sqrt{\text{diag}\left(\Delta I + \sum\limits_{\tau=1}^{t} g_\tau g_\tau^\top\right)}}, \tag{2}$$

where $g_t$ represents an unbiased estimator of $\nabla f(w_t)$, i.e., $\mathbb{E}\left[g_t \mid w_t\right] = \nabla f(w_t)$, $\text{diag}(M) \in \mathbb{R}^d$ is a vector of diagonal elements of matrix $M \in \mathbb{R}^{d \times d}$, $\Delta > 0$, and the division by vector is done component-wise. Ward et al. (2020) has shown that this method achieves a convergence rate of $\mathcal{O}\left(\log T / \sqrt{T}\right)$ for smooth functions, similar to SGD, without prior knowledge of the functions' parameters. However, the performance of AdaGrad deteriorates when applied to data that may exhibit poor scaling or ill-conditioning. In this work, we propose a novel algorithm, KATE, to address the issues of poor data scaling. KATE is also a stochastic adaptive algorithm that can achieve a convergence rate of $\mathcal{O}\left(\log T / \sqrt{T}\right)$ for smooth non-convex functions in terms of $\min_{t \in [T]} \mathbb{E}\left[\|\nabla f(w_t)\|\right]^2$.

## 1.1 Related Work

A significant amount of research has been done on adaptive methods over the years, including AdaGrad (Duchi et al., 2011; McMahan and Streeter, 2010), AMSGrad (Reddi et al., 2019), RMSProp (Tieleman and Hinton, 2012), AI-SARAH (Shi et al., 2023), and Adam (Kingma and Ba, 2014). However, all these works assume that the optimization problem is contained in a bounded set. To address this issue, Li and Orabona (2019) proposes a variant of the AdaGrad algorithm, which does not use the gradient of the last iterate (this makes the step sizes of $t$-th iteration conditionally independent of $g_t$) for computing the step sizes and proves convergence for the unbounded domain.

Each of these works considers a vector of step sizes for each coefficient. Duchi et al. (2011) and McMahan and Streeter (2010) simultaneously proposed the original AdaGrad algorithm. However, McMahan and Streeter (2010) was the first to consider the vanilla scalar form of AdaGrad, known as

$$\text{AdaGradNorm: } w_{t+1} = w_t - \frac{\beta g_t}{\sqrt{\Delta + \sum_{\tau=0}^{t} \|g_\tau\|^2}}. \tag{3}$$

Later, Ward et al. (2020) analyzed AdaGradNorm for minimizing smooth non-convex functions. In a follow-up study, Xie et al. (2020) proves a linear convergence of AdaGradNorm for strongly convex functions. Recently, Liu et al. (2022) analyzed AdaGradNorm for solving smooth convex functions without the bounded domain assumption. Moreover, Liu et al. (2022) extends the convergence guarantees of AdaGradNorm to quasi-convex functions [2] using the function value gap. Orabona et al. (2015) introduce the notion of scale-invariance, which is a special case of affine invariance (Nesterov and Nemirovskii, 1994; Nesterov, 2018; d'Aspremont et al., 2018), propose a scale-invariant version of AdaGrad for online convex optimization for generalized linear models, and prove $\mathcal{O}(\sqrt{T})$ regret bounds in this setup.

Recently, Defazio and Mishchenko (2023) introduced the D-Adaptation method, which has gathered considerable attention due to its promising empirical performances. In order to choose the adaptive step size optimally, one requires knowledge of the initial distance from the solution, i.e., $D := \|w_0 - w_*\|$ where $w_* \in \operatorname{argmin}_{w \in \mathbb{R}^d} f(w)$. The D-Adaptation method works by maintaining an estimate of $D$ and the stepsize choice in this case is $d_t / \sqrt{\sum_{\tau=0}^{t} \|g_\tau\|^2}$ for the $t$-th iteration (here $d_t$ is an estimate of $D$). Mishchenko and Defazio (2023) further modifies the algorithm in a follow-up work and introduces Prodigy (with stepsize choice $d_t^2 / \sqrt{\sum_{\tau=0}^{t} d_\tau^2 \|g_\tau\|^2}$) to improve the convergence speed.

Another exciting line of work on adaptive methods is Polyak stepsizes. Polyak (1969) first proposed Polyak stepsizes for subgradient methods, and recently, the stochastic version (also known as SPS) was introduced by Oberman and Prazeres (2019); Loizou et al. (2021); Abdukhakimov et al. (2024, 2023); Li et al. (2023) and Gower et al. (2021). For a finite sum problem $\min_{w \in \mathbb{R}^d} f(w) := \frac{1}{n} \sum_{i=1}^{n} f_i(w)$, Loizou et al. (2021) uses $\frac{f_i(w_t) - f_i^*}{c \|\nabla f_i(w_t)\|^2}$ as their stepsize choices (here $f_i^* := \min_{w \in \mathbb{R}^d} f_i(w)$), while Oberman and Prazeres (2019) uses $\frac{2(f(w_t) - f^*)}{\mathbb{E}[\|\nabla f_i(w_t)\|^2]}$ for $k$-th iteration. However, these methods are impractical when $f^*$ or $f_i^*$ is unknown. Following its introduction,

---

[2] $f$ satisfy $f^* \geq f(w) + \frac{1}{\zeta} \langle f(w), w^* - w \rangle$ for some $\zeta \in (0, 1]$ where $w^* \in \operatorname{argmin}_w f(w)$.

Table 1: Summary of convergence guarantees for closely-related adaptive algorithms to solve *smooth non-convex stochastic* optimization problems. Convergence rates are given in terms of $\min_{t \in [T]} \mathbb{E} \left[ \|\nabla f(w_t)\| \right]^2$. We highlight KATE's *scale-invariance* property for problems of type (4).

| Algorithm | Convergence rate | Scale invariance |
|---|---|---|
| AdaGradNorm (Ward et al., 2020) | $\mathcal{O}\left( \log T / \sqrt{T} \right)$ | ✗ |
| AdaGrad (Défossez et al., 2020) | $\mathcal{O}\left( \log T / \sqrt{T} \right)$ | ✗ |
| Adam (Défossez et al., 2020) | $\mathcal{O}\left( \log T / \sqrt{T} \right)$ | ✗ |
| KATE (this work) | $\mathcal{O}\left( \log T / \sqrt{T} \right)$ | ✓ |

several variants of the SPS algorithm emerged (Li et al., 2023; D'Orazio et al., 2021). Lately, Orvieto et al. (2022) tackled the issues with unknown $f_i^*$ and developed a truly adaptive variant. In practice, the SPS method shows excellent empirical performance on overparameterized deep learning models (which satisfy the interpolation condition i.e. $f_i^* = 0, \ \forall i \in [n]$) (Loizou et al., 2021).

### 1.2 Main Contribution

Our main contributions are summarized below.

• KATE**: new scale-invariant version of** AdaGrad. We propose a new method called KATE that can be seen as a version of AdaGrad, which does not use a square root in the denominator of the stepsize. To compensate for this change, we introduce a new sequence defining the numerator of the stepsize. We prove that KATE is scale-invariant for generalized linear models: if the starting point is zero, then the loss values (and training and test accuracies in the case of classification) at points generated by KATE are independent of the data scaling (Proposition 2.1), meaning that the speed of convergence of KATE is the same as for the best scaling of the data.

• **Convergence for smooth non-convex problems.** We prove that for smooth non-convex problems with noise having bounded variance KATE has $\mathcal{O}(\log(T)/\sqrt{T})$ convergence rate (Theorem 3.4), matching the best-known rates for AdaGrad and Adam (Défossez et al., 2020).

• **Numerical experiments.** We empirically illustrate the scale-invariance of KATE on the logistic regression task and test its performance on logistic regression (see Section 4.1), image classification, and text classification problems (see Section 4.2). In all the considered scenarios, KATE outperforms AdaGrad and works either better or comparable to Adam.

### 1.3 Notation

We denote the set $\{1, 2, \cdots, n\}$ as $[n]$. For a vector $a \in \mathbb{R}^d$, $a[k]$ is the $k$-th coordinate of $a$ and $a^2$ represents the element-wise square of $a$, i.e., $a^2[k] = (a[k])^2$. For two vectors $a$ and $b$, $\frac{a}{b}$ stands for element-wise division of $a$ and $b$, i.e., $k$-th coordinate of $\frac{a}{b}$ is $\frac{a[k]}{b[k]}$. Given a function $h : \mathbb{R}^d \to \mathbb{R}$, we use $\nabla h \in \mathbb{R}^d$ to denote its gradient and $\nabla_k h$ to indicate the $k$-th component of $\nabla h$. Throughout the paper $\| \cdot \|$ represents the $\ell_2$-norm and $f_* = \inf_{w \in \mathbb{R}^d} f(w)$. Moreover, we use $\|w\|_A$ for a positive-definite matrix $A$ to define $\|w\|_A := \sqrt{w^\top A w}$. Furthermore, $\mathbb{E}[\cdot]$ denotes the total expectation while $\mathbb{E}_t[\cdot]$ denotes the conditional expectation conditioned on all iterates up to step $t$ i.e. $w_0, w_1, \ldots, w_t$.

## 2 Motivation and Algorithm Design

We focus on solving the minimization problem (1) using a variant of AdaGrad. We aim to design an algorithm that performs well, irrespective of how poorly the data is scaled.

**Generalized linear models.** Here, we consider the parameter estimation problem in generalized linear models (GLMs) (Nelder and Wedderburn, 1972; Agresti, 2015) using maximum likelihood estimation. GLMs are an extension of linear models and encompass several other valuable models, such as logistic (Hosmer Jr et al., 2013) and Poisson regression (Frome, 1983), as special cases. The parameter estimation to fit GLM on dataset $\{x_i, y_i\}_{i=1}^n$ (where $x_i \in \mathbb{R}^d$ are feature vectors and $y_i$ are response variables) can be reformulated as

$$\min_{w \in \mathbb{R}^d} f(w) := \frac{1}{n} \sum_{i=1}^n \varphi_i \left( x_i^\top w \right) \tag{4}$$

for differentiable functions $\varphi_i : \mathbb{R} \to \mathbb{R}$ (Shalev-Shwartz and Ben-David, 2014; Nguyen et al., 2017b; Takáč et al., 2013; He et al., 2018; Chezhegov et al., 2024). For example, the linear regression on data $\{x_i, y_i\}_{i=1}^n$ is equivalent to solving (4) with $\varphi_i(z) = (z - y_i)^2$. Next, the choice of $\varphi_i$ for logistic regression is $\varphi_i(z) = \log (1 + \exp (-y_i z))$.

**Scale-invariance.** Now consider the instances of fitting GLMs on two datasets $\{x_i, y_i\}_{i=1}^n$ and $\{V x_i, y_i\}_{i=1}^n$, where $V \in \mathbb{R}^{d \times d}$ is a diagonal matrix with positive entries. Note that the second dataset is a scaled version of the first one where the $k$-th component of feature vectors $x_i$ are multiplied by a scalar $V_{kk}$. Then, the minimization problems corresponding to datasets $\{x_i, y_i\}_{i=1}^n$ and $\{V x_i, y_i\}_{i=1}^n$ are (4) and

$$\min_{w \in \mathbb{R}^d} f^V(w) := \frac{1}{n} \sum_{i=1}^n \varphi_i \left( x_i^\top V w \right), \tag{5}$$

respectively, for functions $\varphi_i$. In this work, *we want to design an algorithm with equivalent performance for the problems* (4) *and* (5). If we can do that, the new algorithm's performance will not deteriorate for poorly scaled data, i.e., the method will be scale-invariant (Orabona et al., 2015), which is a special case of affine-invariance, see (Nesterov and Nemirovskii, 1994; Nesterov, 2018; d'Aspremont et al., 2018). To develop such an algorithm, we replace the denominator of AdaGrad step size with its square (remove the square root from the denominator), i.e., $\forall k \in [d]$

$$w_{t+1}[k] = w_t[k] - \frac{\beta m_t[k]}{\sum_{\tau=0}^t g_\tau^2[k]} g_t[k] \tag{6}$$

for some $m_t \in \mathbb{R}^d$.[3] The following proposition shows that this method (6) satisfies a scale-invariance property with respect to functional value.

**Proposition 2.1** (Scale invariance). Suppose we solve problems (4) and (5) using algorithm (6). Then, the iterates $\hat{w}_t$ and $\hat{w}_t^V$ corresponding to (4) and (5) follow: $\forall k \in [d]$

$$\hat{w}_{t+1}[k] = \hat{w}_t[k] - \frac{\beta m_t[k]}{\sum_{\tau=0}^t g_\tau^2[k]} g_t[k], \tag{7}$$

$$\hat{w}_{t+1}^V[k] = \hat{w}_t^V[k] - \frac{\beta m_t[k]}{\sum_{\tau=0}^t \left( g_\tau^V[k] \right)^2} g_t^V[k] \tag{8}$$

with $g_\tau = \varphi_{i_\tau}'(x_{i_\tau}^\top \hat{w}_\tau) x_{i_\tau}$ and $g_\tau^V = \varphi_{i_\tau}'(x_{i_\tau}^\top V \hat{w}_\tau) V x_{i_\tau}$ for $i_\tau$ chosen uniformly from $[n]$, $\tau = 0, 1, \ldots, t$, $t \geq 0$. Moreover, updates (7) and (8) satisfy

$$\hat{w}_t = V \hat{w}_t^V, \quad V g_t = g_t^V, \quad f(\hat{w}_t) = f^V(\hat{w}_t^V) \tag{9}$$

for all $t \geq 0$ when $\hat{w}_0 = \hat{w}_0^V = 0 \in \mathbb{R}^d$. Furthermore we have

$$\left\| g_t^V \right\|_{V^{-2}}^2 = \left\| g_t \right\|^2. \tag{10}$$

The Proposition 2.1 highlights that the update rule of the form (6) satisfies a scale-invariance property for GLMs. In contrast, AdaGrad does not satisfy (9) and (10). In Appendix C, we illustrate numerically the scale-invariance of KATE and the lack of the scale-invariance of AdaGrad. We also emphasize that AdaGrad with $\Delta = 0$ is known to be a scale-free method[4].

---

[3]Sequence $\{m_t\}_{t \geq 0}$ can depend on the problem but is assumed to be scale-invariant.

[4]The algorithm is called scale-free if for any $c > 0$, it generates the same sequence of points for functions $f$ and $cf$ given the same initialization and hyperparameters. To the best of our knowledge, this definition is

---

**Algorithm 1** KATE

---

**Require:** Initial point $w_0 \in \mathbb{R}^d$, step size $\beta > 0, \eta \in \mathbb{R}^d_+$ and $b_{-1}, m_{-1} = 0$.

1: **for** $t = 0, 1, ..., T$ **do**
2:      Compute $g_t \in \mathbb{R}^d$ such that $\mathbb{E}[g_t] = \nabla f(w_t)$.
3:      $b_t^2 = b_{t-1}^2 + g_t^2$
4:      $m_t^2 = m_{t-1}^2 + \eta g_t^2 + \frac{g_t^2}{b_t^2}$
5:      $w_{t+1} = w_t - \frac{\beta m_t}{b_t^2} g_t$

---

**Design of** KATE**.** In order to construct an algorithm following the update rule (6), one may choose $m_t[k] = 1 \ \forall k \in [d]$. However, the step size from (6) in this case may decrease very fast, and the resulting method does not necessarily converge. Therefore, we need a more aggressive choice of $m_t$, which grows with $t$. It motivates the construction of our algorithm KATE (Algorithm 1),[5] where we choose $m_t[k] = \sqrt{\eta[k]b_t^2[k] + \sum_{\tau=0}^{t} \frac{g_\tau^2[k]}{b_\tau^2[k]}}$. Note that the term $\sum_{\tau=0}^{t} \frac{g_\tau^2[k]}{b_\tau^2[k]}$ is scale-invariant for GLMs (follows from Proposition 2.1). To make $m_t$ scale-invariant, we choose $\eta \in \mathbb{R}^d$ in the following way:

- $\eta \to 0$: When $\eta$ is very small, $m_t$ is also approximately scale-invariant for GLMs.
- $\eta = 1/(\nabla f(w_0))^2$: In this case $\eta b_t^2 = b_t^2/(\nabla f(w_0))^2$ is scale-invariant for GLMs (follows from Proposition 2.1) as well as $m_t$.

KATE can be rewritten in the following coordinate form

$$w_{t+1}[k] = w_t[k] - \nu_t[k]g_t[k], \qquad \forall k \in [d], \tag{11}$$

where $g_t$ is an unbiased estimator of $\nabla f(w_t)$ and the per-coefficient step size $\nu_t[k]$ is defined as

$$\nu_t[k] := \frac{\beta\sqrt{\eta[k]b_t^2[k] + \sum_{\tau=0}^{t} \frac{g_\tau^2[k]}{b_\tau^2[k]}}}{b_t^2[k]}. \tag{12}$$

Note that the numerator of the steps $\nu_t[k]$ is increasing with iterations $t$. However, one of the crucial properties of this step size choice is that the steps always decrease with $t$, which we rely on in our convergence analysis.

> **Lemma 2.2** (Decreasing step size). For $\nu_t[k]$ defined in (11) we have
>
> $$\nu_{t+1}[k] \leq \nu_t[k], \qquad \forall k \in [d]. \tag{13}$$

**Comparison with the scale-invariant version of** AdaGrad **by** Orabona et al. (**2015**)**.** In the special case of GLMs, Orabona et al. (2015) propose a different version of AdaGrad. The method is proposed for the case of online convex optimization, and in the case of standard optimization with GLMs (4), it has the following form

$$w_0 := 0, \quad w_{t+1} := -\beta \frac{\sum_{\tau=0}^{t} \nabla f_{i_\tau}(w_\tau)}{a_t^2\sqrt{d}\sqrt{\gamma^2 + \sum_{\tau=0}^{t}\left(\nabla f_{i_\tau}(w_\tau)/a_\tau\right)^2}}, \quad a_t := \max_{\tau=0,...,t}|x_{i_\tau}|, \tag{14}$$

where $\{i_\tau\}_{\tau=0}^{t}$ are arbitrary indices from $[n]$ (e.g., selected uniformly at random), functions $f_i : \mathbb{R}^d \to \mathbb{R}$ are defined as $f_i(w) := \varphi_i(x_i^\top w)$ for $i \in [n]$, and $\gamma$ is such that $f_i(w)$ is $\gamma$-Lipschitz for $i \in [n]$. In this setup, the update rule of KATE with $w_0 = 0$ can be written as follows:

$$w_{t+1} := -\beta \sum_{\tau=0}^{t} \frac{m_\tau}{b_\tau^2}\nabla f_{i_\tau}(w_\tau), \quad m_t := \sqrt{\eta \sum_{\tau=0}^{t}(\nabla f_{i_\tau}(w_\tau))^2 + \sum_{\tau=0}^{t}(\nabla f_{i_\tau}(w_\tau))^2/b_\tau^2},$$

---

introduced by Cesa-Bianchi et al. (2005, 2007) in the context of learning with expert advice and extended to the context of generic online convex optimization by Orabona and Pál (2015, 2018). We emphasize that scale-freeness and scale-invariance are completely different concepts.

[5]Note that, for $m_t = b_t \forall t$ we get the AdaGrad algorithm.

where $b_t := \sqrt{\sum_{\tau=0}^{t} (\nabla f_{i_\tau}(w_\tau))^2}$, $\{i_\tau\}_{\tau=0}^{t}$ are sampled from $[n]$ uniformly at random. Although both methods can be seen as variations of AdaGrad due to the terms $\sum_{\tau=0}^{t} \left(\nabla f_{i_\tau}(w_\tau/a_\tau)\right)^2$ and $\sum_{\tau=0}^{t} \left(\nabla f_{i_\tau}(w_\tau)\right)^2$ respectively, the scale-invariance is achieved quite differently in these methods. The method from (14) uses the feature vectors explicitly in the update rule to ensure scale-invariance: indeed, the square root in the definition of $w_{t+1}$ is independent of scaling, and $a_t^2$ in the denominator ensures that $\hat{w}_{t+1} = V\hat{w}_{t+1}^V$ if we define them similarly to KATE (see equations (7)-(8)). In contrast, KATE achieves the scale-invariance by removing the square root from the denominator (as explained earlier). Moreover, unlike the method from (14), KATE does not use the feature vectors explicitly in its update rule (only in the gradients of $f_{i_\tau}$) and, thus, can be used for general stochastic optimization (not necessarily for the case of GLMs).

## 3 Convergence Analysis

In this section, we present and discuss the convergence guarantees of KATE. In the first subsection, we list the assumptions made about the problem.

### 3.1 Assumptions

In all our theoretical results, we assume that $f$ is smooth as defined below.

**Assumption 3.1** ($L$-smooth). Function $f$ is $L$-smooth, i.e. for all $w, w' \in \mathbb{R}^d$

$$f(w') \leq f(w) + \langle \nabla f(w), w' - w \rangle + \frac{L}{2} \|w - w'\|^2. \tag{15}$$

This assumption is standard in the literature of adaptive methods (Li and Orabona, 2019; Ward et al., 2020; Liu et al., 2022; Nguyen et al., 2018, 2021, 2017a; Beznosikov and Takáč, 2021). Moreover, we assume that at any iteration $t$ of KATE, we can access $g_t$ — a noisy and unbiased estimate of $\nabla f(w_t)$. We also make the following assumption on the noise of the gradient estimate $g_t$.

**Assumption 3.2** (Bounded Variance). For fixed constant $\sigma > 0$, the variance of the stochastic gradient $g_t$ (unbiased estimate of $\nabla f(w_t)$) at any time $t$ satisfies

$$\mathbb{E}_t \left[ \|g_t - \nabla f(w_t)\|^2 \right] \leq \sigma^2. \tag{BV}$$

Bounded variance is a common assumption to study the convergence of stochastic gradient-based methods. Several assumptions on stochastic gradients are used in the literature to explore the adaptive methods. Ward et al. (2020) used the BV, while Liu et al. (2022) assumed the sub-Weibull noise, i.e. $\mathbb{E}\left[ \exp\left( \|g_t - \nabla f(w_t)\|/\sigma \right)^{1/\theta} \right] \leq \exp(1)$ for some $\theta > 0$, to prove the convergence of AdaGradNorm. Li and Orabona (2019) assumes sub-Gaussian ($\theta = 1/2$ in sub-Weibull condition) noise to study a variant of AdaGrad. However, sub-Gaussian noise is strictly stronger than BV. Recently, Faw et al. (2022) analyzed AdaGradNorm under a more relaxed condition known as affine variance $\left( \text{i.e. } \mathbb{E}_t \left[ \|g_t - \nabla f(w_t)\|^2 \right] \leq \sigma_0^2 + \sigma_1^2 \|\nabla f(w_t)\|^2 \right)$.

### 3.2 Main Results

In this section, we cover the main convergence guarantees of KATE for both deterministic and stochastic setups.

**Deterministic setting.** We first present our results for the deterministic setting. In this setting, we consider the gradient estimate to have no noise (i.e. $\sigma^2 = 0$) and $g_t = \nabla f(w_t)$. The main result in this setting is summarized below.

**Theorem 3.3.** Suppose $f$ satisfy Assumption 3.1 and $g_t = \nabla f(w_t)$. Moreover, $\beta > 0$ and $\eta[k] > 0$ are chosen such that $\nu_0[k] \leq \frac{1}{L}$ for all $k \in [d]$. Then the iterates of KATE satisfies

$$\min_{t \leq T} \|\nabla f(w_t)\|^2 \leq \frac{\left(\frac{2(f(w_0) - f_*)}{\sqrt{\eta_0}\beta} + \sum_{k=1}^{d} b_0[k]\right)^2}{T+1},$$

where $\eta_0 \coloneqq \min_{k \in [d]} \eta[k]$.

**Discussion on Theorem 3.3.** Theorem 3.3 establishes an $\mathcal{O}\left(1/T\right)$ convergence rate for KATE, which is optimal for finding a first-order stationary point of a non-convex problem (Carmon et al., 2020). However, this result is not parameter-free. To prove the convergence, we assume that $\nu_0[k] \leq \frac{1}{L}$, $\forall k \in [d]$ in Theorem 3.3, which is equivalent to $\beta\sqrt{1 + \eta_0 \left(\nabla_k f(w_0)\right)^2} \leq \left(\nabla_k f(w_0)\right)^2/L$, $\forall k \in [d]$. Note that the later condition holds for sufficiently small (dependent on $L$) values of $\beta, \eta_0 > 0$.

However, it is possible to derive a parameter-free version of Theorem 3.3. Indeed, Lemma 2.2 implies that the step sizes are decreasing. Therefore, we can break down the analysis of KATE into two phases: Phase I when $\nu_0[k] > 1/L$ and Phase II when $\nu_0[k] \leq 1/L$, when the current analysis works, and then follow the proof techniques of Ward et al. (2020) and Xie et al. (2020). We leave this extension as a possible future direction of our work.

**Stochastic setting.** Next, we present the convergence guarantees for KATE in the stochastic case, when we can access an unbiased gradient estimate $g_t$ with non-zero noise.

**Theorem 3.4.** Suppose $f$ satisfy Assumption 3.1 and $g_t$ is an unbiased estimator of $\nabla f(w_t)$ such that BV holds. Moreover, we assume $\|\nabla f(w_t)\|^2 \leq \gamma^2$ for all $t$. Then the iterates of KATE satisfy

$$\min_{t \leq T} \mathbb{E}\left[\|\nabla f(w_t)\|\right] \leq \left(\frac{\|g_0\|}{T} + \frac{2(\gamma + \sigma)}{\sqrt{T}}\right)^{1/2} \sqrt{\frac{2\mathcal{C}_f}{\beta\sqrt{\eta_0}}},$$

where $\eta_0 \coloneqq \min_{k \in [d]} \eta[k]$ and

$$
\begin{aligned}
\mathcal{C}_f \coloneqq \ & f(w_0) - f_* + 2\beta\sigma\sum_{k=1}^{d}\sqrt{\eta[k]}\log\left(\frac{e(\sigma^2 + \gamma^2)T}{g_0^2[k]}\right) \\
& + \sum_{k=1}^{d}\left(\frac{\beta^2\eta[k]L}{2} + \frac{\beta^2 L}{2g_0^2[k]}\right)\log\left(\frac{e(\sigma^2 + \gamma^2)T}{g_0^2[k]}\right).
\end{aligned}
$$

**Comparison with prior work.** Theorem 3.4 shows an $\mathcal{O}(\log^{1/2} T/T^{1/4})$ convergence rate for KATE with respect to the metric $\min_{t \leq T} \mathbb{E}\left[\|\nabla f(w_t)\|\right]$ for the stochastic setting. Note that, in the stochastic setting, KATE achieves a slower rate than Theorem 3.3 due to noise accumulation. Up to the logarithmic factor, this rate is optimal (Arjevani et al., 2023). Similar rates for the same metric follow from the results[6] of (Défossez et al., 2020) for AdaGrad and Adam.

Finally, Li and Orabona (2019) considers a variant of AdaGrad closely related to KATE:

$$w_{t+1} = w_t - \frac{\beta g_t}{\left(\text{diag}\left(\Delta I + \sum_{\tau=1}^{t-1} g_\tau g_\tau^\top\right)\right)^{\frac{1}{2}+\varepsilon}}, \tag{16}$$

for some $\varepsilon \in [0, 1/2)$ and $\Delta > 0$. It differs from AdaGrad in two key aspects: the denominator of the stepsize does not contain the last stochastic gradient, and also, instead of the square root of the sum of squared gradients, this sum is taken in the power of $1/2 + \varepsilon$. However, the results from Li and Orabona (2019) do not imply convergence for the case of $\varepsilon = 1/2$, which is expected since, in this case, the stepsize converges to zero too quickly in general. To compensate for such a rapid decrease, in KATE, we introduce an increasing sequence $m_t$ in the numerator of the stepsize.

---

[6]Défossez et al. (2020) derive $\mathcal{O}(\log T/\sqrt{T})$ convergence rates for AdaGrad and Adam in terms of $\min_{t \leq T} \mathbb{E}\left[\|\nabla f(w_t)\|^2\right]$ which is not smaller than $\min_{t \leq T}\left(\mathbb{E}\left[\|\nabla f(w_t)\|\right]\right)^2$.

**Proof technique.** Compared to the AdaGrad, KATE uses more aggressive steps (the larger numerator of KATE due to the extra term $\sum_{\tau=0}^{t} g_\tau^2[k]/b_\tau^2[k]$). Therefore, we expect KATE to have better empirical performance. However, introducing $\sum_{\tau=0}^{t} g_\tau^2[k]/b_\tau^2[k]$ in the numerator raises additional technical difficulties in the proof technique. Fortunately, as we rigorously show, the KATE steps $\nu_t[k]$ retain some of the critical properties of AdaGrad steps. For instance, they (i) are lower bounded by AdaGrad steps up to a constant, (ii) decrease with iteration $t$ (Lemma 2.2), and (iii) have closed-form upper bounds for $\sum_{t=0}^{T} \nu_t^2[k]g_t^2[k]$. These are indeed the primary building blocks of our proof technique.

## 4   Numerical Experiments

In this section, we implement KATE in several machine learning tasks to evaluate its performance. To ensure transparency and facilitate reproducibility, we provide an access to the source code for all of our experiments at https://github.com/nazya/KATE.

### 4.1   Logistic Regression

In this section, we consider the logistic regression model

$$\min_{w \in \mathbb{R}^d} f(w) = \frac{1}{n} \sum_{i=1}^{n} \log\left(1 + \exp\left(-y_i x_i^\top w\right)\right), \tag{17}$$

to elaborate on the scale-invariance and robustness of KATE for various initializations. For the experiments of this Section 4.1, we used Mac mini (M1, 2020), RAM 8 GB and storage 256 GB. Each of these plots took about 20 minutes to run.

#### 4.1.1   Robustness of KATE

To conduct this experiment, we set the total number of samples to 1000 (i.e. $n = 1000$). Here, we simulate the independent vectors $x_i \in \mathbb{R}^{20}$ such that each entry is from $\mathcal{N}(0, 1)$. Moreover, we generate a diagonal matrix $V \in \mathbb{R}^{20 \times 20}$ such that $\log V_{kk} \stackrel{\text{iid}}{\sim} \text{Unif}(-10, 10)$, $\forall k \in [20]$. Similarly, we generate $w^* \in \mathbb{R}^{20}$ with each component from $\mathcal{N}(0, 1)$ and set the labels

$$y_i = \begin{cases} 1, & x_i^\top V w^* \geq 0, \\ -1, & x_i^\top V w^* < 0, \end{cases} \qquad \forall i \in [n].$$

We compare KATE's performance with four other algorithms: AdaGrad, AdaGradNorm, SGD-decay and SGD-constant, similar to the section 5.1 of Ward et al. (2020). For each algorithm, we initialize with $w_0 = 0 \in \mathbb{R}^{20}$ and independently draw a sample of mini-batch size 10 to update the weight vector $w_t$. We compare the algorithms • AdaGrad with stepsize $\frac{\beta}{\sqrt{\Delta + \sum_{\tau=0}^{t} g_\tau^2}}$, • AdaGradNorm with step size $\frac{\beta}{\sqrt{\Delta + \sum_{\tau=0}^{t} \|g_\tau\|^2}}$, • SGD-decay with stepsize $\beta/\Delta\sqrt{t+1}$, and • SGD-constant with step size $\beta/\Delta$. Similarly, for KATE we use stepsize $\frac{\beta m_t}{b_t^2}$ where $m_t^2 = \eta b_t^2 + \sum_{\tau=0}^{t} g_\tau^2/b_\tau^2$ and $b_t^2 = \Delta + \sum_{\tau=0}^{t} g_\tau^2$. Here, we choose $\beta = f(w_0) - f(w^*)$ and vary $\Delta$ in $\{10^{-8}, 10^{-6}, 10^{-4}, 10^{-2}, 1, 10^2, 10^4, 10^6, 10^8\}$.

In Figures 1a, 1b, and 1c, we plot the functional value $f(w_t)$ (on the $y$-axis) after $10^4, 5 \times 10^4$, and $10^5$ iterations, respectively. In theory, the convergence of SGD requires the knowledge of smoothness constant $L$. Therefore, when the $\Delta$ is small (hence the stepsize is large), SGD-decay and SGD-constant diverge. However, the adaptive algorithms KATE, AdaGrad, and AdaGradNorm can auto-tune themselves and converge for a wide range of $\Delta$s (even when the $\Delta$ is too small). As we observe in Figure 1, when the $\Delta$ is small, KATE outperforms all other algorithms. For instance, when $\Delta = 10^{-8}$, KATE achieves a functional value of $10^{-3}$ after only $10^4$ iterations (see Figure 1a), while other algorithms fail to achieve this even after $10^5$ iterations (see Figure 1c). Furthermore, KATE performs as well as AdaGrad and better than other algorithms when the $\Delta$ is large. *In particular, this experiment highlights that KATE is robust to initialization $\Delta$.*

#### 4.1.2   Peformance of KATE on Real Data

In this section, we examine KATE's performance on real data. We test KATE on three datasets: heart, australian, and splice from the LIBSVM library (Chang and Lin, 2011). The response variables $y_i$ of

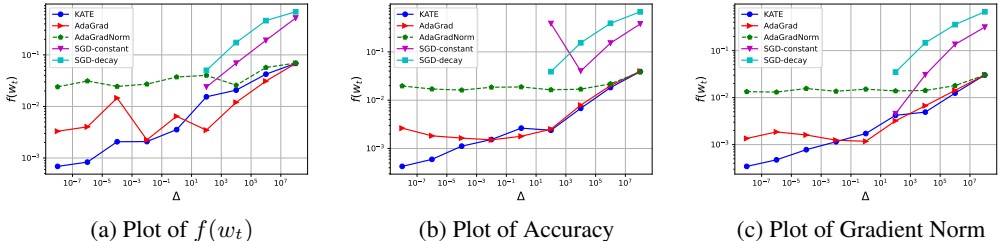

(a) Plot of $f(w_t)$         (b) Plot of Accuracy         (c) Plot of Gradient Norm

Figure 1: Comparison of KATE with AdaGrad, AdaGradNorm, SGD-decay and SGD-constant for different values of $\Delta$ (on $x$-axis for logistic regression model. Figure 1a, 1b and 1c plots the functional value $f(w_t)$ (on $y$-axis) after $10^4, 5 \times 10^4$, and $10^5$ iterations respectively.

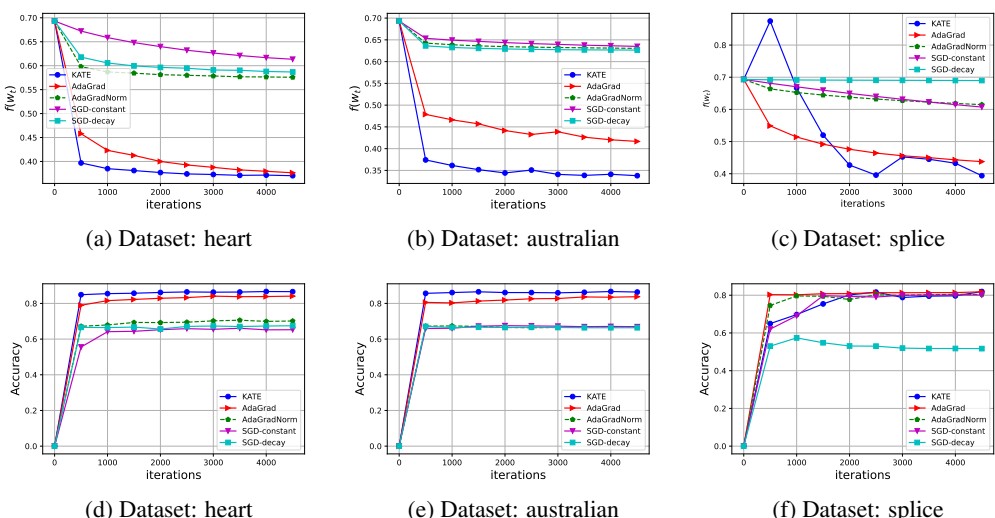

(a) Dataset: heart         (b) Dataset: australian         (c) Dataset: splice

(d) Dataset: heart         (e) Dataset: australian         (f) Dataset: splice

Figure 2: Comparison of KATE with AdaGrad, AdaGradNorm, SGD-decay and SGD-constant on datasets heart, australian, and splice from LIBSVM. Figures 2a, 2b and 2c plot the functional value $f(w_t)$, while 2d, 2e and 2f plot the accuracy on $y$-axis for $5,000$ iterations.

each of these datasets contain two classes, and we use them for binary classification tasks using a logistic regression model (17). We take $\eta = 1/(\nabla f(w_0))^2$ for KATE and tune $\beta$ in all the experiments. For tuning $\beta$, we do a grid search on the list $\{10^{-10}, 10^{-8}, 10^{-6}, 10^{-4}, 10^{-2}, 1\}$. Similarly, we tune stepsizes for other algorithms. We take 5 trials for each of these algorithms and plot the mean of their trajectories.

We plot the functional value $f(w_t)$ (i.e. loss function) in Figures 2a, 2b and 2c, whereas Figures 2d, 2e and 2f plot the corresponding accuracy of the weight vector $w_t$ on the $y$-axis for $5,000$ iterations. We observe that KATE performs superior to all other algorithms, even on real datasets.

### 4.2 Training of Neural Networks

In this section, we compare the performance of KATE, AdaGrad and Adam on two tasks, i.e. training ResNet18 (He et al., 2016) on the CIFAR10 dataset (Krizhevsky and Hinton, 2009) and BERT (Devlin et al., 2018) fine-tuning on the emotions dataset (Saravia et al., 2018) from the Hugging Face Hub. We use internal cluster with the following hardware: AMD EPYC 7552 48-Core Processor, 512GiB RAM, NVIDIA A100 40GB GPU, 200gb user storage space.

**General comparison.** We choose standard parameters for Adam ($\beta_1 = 0.9$ and $\beta_2 = 0.999$) that are default values in PyTorch and select the learning rate of $10^{-5}$ for all considered methods. We run KATE with different values of $\eta \in \{0, 10^{-1}, 10^{-2}\}$. For the image classification task, we normalize the images (similar to Horváth and Richtárik (2020)) and use a mini-batch size of 500. For the BERT fine-tuning, we use a mini-batch size 160 for all methods.

Figures 3-8 report the evolution of top-1 accuracy and cross-entropy loss (on the $y$-axis) calculated on the test data. For the image classification task, we observe that KATE with different choices of

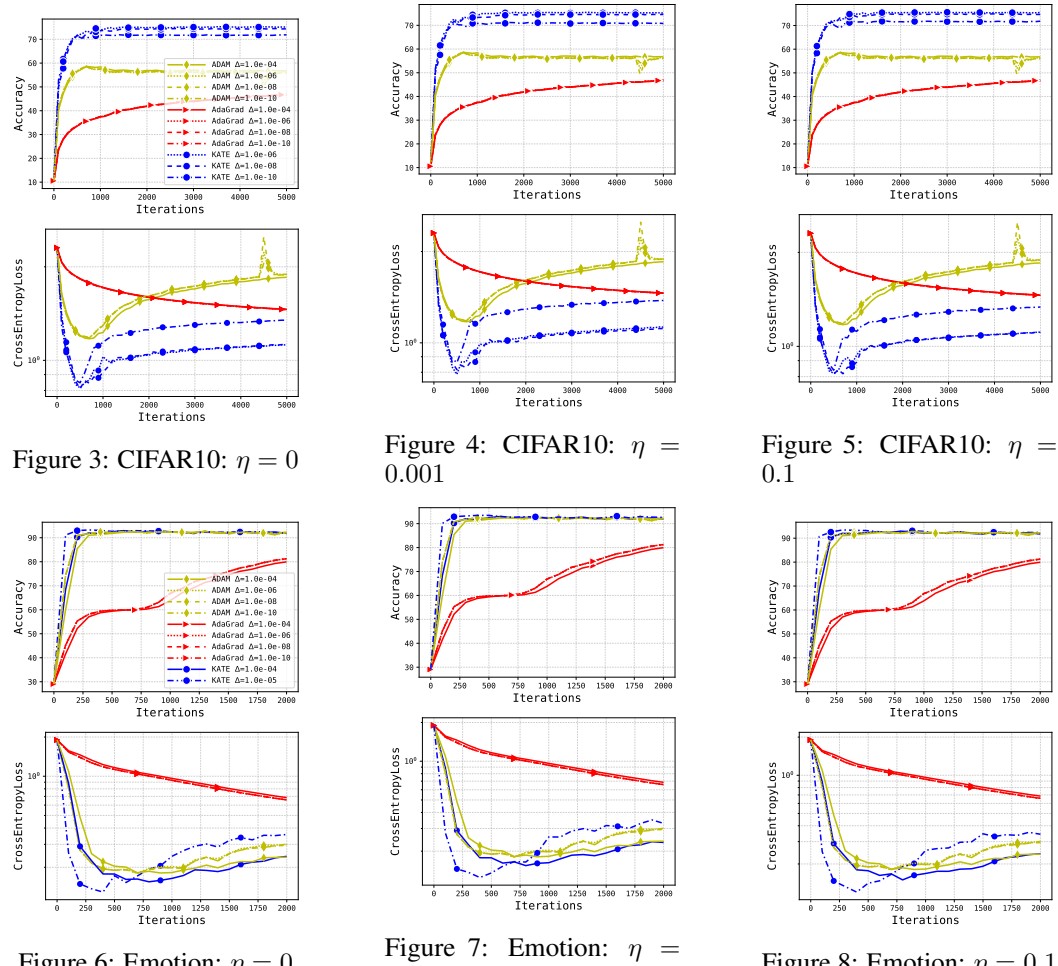

Figure 3: CIFAR10: $\eta = 0$

Figure 4: CIFAR10: $\eta = 0.001$

Figure 5: CIFAR10: $\eta = 0.1$

Figure 6: Emotion: $\eta = 0$

Figure 7: Emotion: $\eta = 0.001$

Figure 8: Emotion: $\eta = 0.1$

$\eta$ outperforms Adam and AdaGrad. Finally, we also observe that KATE performs comparably to Adam on the BERT fine-tuning task and is better than AdaGrad. These preliminary results highlight the potential of KATE to be applied for training neural networks for different tasks. For BERT each run takes about 35 minutes, and 25 minutes for ResNet.

**Hyper-parameters tuning.** Next, we compare baselines presented in Saravia et al. (2018) for emotions classification and Zhang et al. (2019) for image classification. These papers provide efficient setups for learning rates and learning rate schedulers that are reasonable to compare with. Saravia et al. (2018) performs a search of efficient learning rate and uses a linear learning rate scheduler with warmup for Adam optimizer. A different learning rate (1e-5), $\Delta$=1e-5 and the same scheduler applied for KATE lead to the same performance, see Figure 9. We would like to point out that it is challenging to find a reference for hyper-parameters for a certain setup. Thus, to fairly compare with Saravia et al. (2018) we use distilroberta-base model. Zhang et al. (2019) did a grid search for an efficient learning rate and used a multi-step scheduler for Adam optimizer, decaying the learning rate by a factor of 5 at the 60th, 120th, and 160th epochs. Zhang et al. (2019) refers to DeVries and Taylor (2017) for the code implementing special techniques, namely data augmentation and cutout to achieve higher accuracy. A different learning rate (1e-3), the same scheduler and $\Delta$=1e-3 applied for KATE demonstrates comparable performance, see Figure 10. For BERT each run takes about 20 minutes, while 100 minutes for ResNet.

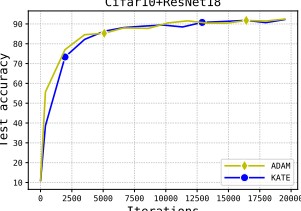

Figure 9: Cifar10: $\eta = 0.001$

Figure 10: Emotion: $\eta = 0.001$

## Acknowledgments and Disclosure of Funding

We thank Francesco Orabona and Dmitry Kamzolov for the pointers to the related works that we missed while preparing the first version of this paper. We also thank anonymous reviewers for their useful feedback and suggestions.

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

# Supplementary Material

## Contents

# A Technical Lemmas

**Lemma A.1** (AM-GM). For $\lambda > 0$ we have

$$ab \leq \frac{\lambda}{2}a^2 + \frac{1}{2\lambda}b^2. \tag{18}$$

**Lemma A.2** (Cauchy-Schwarz Inequality). For $a_1, \cdots, a_n, b_1, \cdots, b_n \in \mathbb{R}$ we have

$$\left(\sum_{i=1}^n a_i^2\right)\left(\sum_{i=1}^n b_i^2\right) \geq \left(\sum_{i=1}^n a_i b_i\right)^2. \tag{19}$$

**Lemma A.3** (Holder's Inequality). Suppose $X, Y$ are two random variables and $p, q > 1$ satisfy $\frac{1}{p} + \frac{1}{q} = 1$. Then

$$\mathbb{E}\left(|XY|\right) \leq \left(\mathbb{E}\left(|X|^p\right)\right)^{\frac{1}{p}} \left(\mathbb{E}\left(|Y|^q\right)\right)^{\frac{1}{q}}. \tag{20}$$

**Lemma A.4** (Jensen's Inequality). For a convex function $g : \mathbb{R}^d \to \mathbb{R}$ and a random variable $X$ such that $\mathbb{E}(\Psi(X))$ and $\Psi(\mathbb{E}(X))$ are finite, we have

$$\Psi\left(\mathbb{E}(X)\right) \leq \mathbb{E}(\Psi(X)). \tag{21}$$

**Lemma A.5.** For $a_1, a_2, \cdots, a_n \geq 0$ and $b_1, b_2, \cdots, b_n > 0$ we have

$$\sum_{i=1}^n \frac{a_i}{\sqrt{b_i}} \geq \frac{\sum_{i=1}^n a_i}{\sqrt{\sum_{i=1}^n b_i}}. \tag{22}$$

*Proof.* Expanding the LHS of (22) we get

$$\left(\sum_{i=1}^n \frac{a_i}{\sqrt{b_i}}\right)^2 = \sum_{i=1}^n \frac{a_i^2}{b_i} + 2\sum_{i \neq j} \frac{a_i a_j}{\sqrt{b_i b_j}}$$

$$\geq \sum_{i=1}^n \frac{a_i^2}{b_i}. \tag{23}$$

The last inequality follows from $\frac{a_i}{\sqrt{b_i}} \geq 0$ for all $i \in [n]$. Now, using Cauchy-Schwarz Inequality (19), we have

$$\left(\sum_{i=1}^n \frac{a_i^2}{b_i}\right)\left(\sum_{i=1}^n b_i\right) \geq \left(\sum_{i=1}^n a_i\right)^2. \tag{24}$$

Then combining (23) and (24), we get

$$\left(\sum_{i=1}^n \frac{a_i}{\sqrt{b_i}}\right)^2 \left(\sum_{i=1}^n b_i\right) \geq \left(\sum_{i=1}^n a_i\right)^2.$$

Finally dividing both sides by $\sum_{i=1}^n b_i$ and taking square root we get the desired result. $\square$

**Lemma A.6.** For $k \in [d]$ and $t \geq 1$ we have

$$
\mathbb{E}_t\left[\left(\frac{\beta\sqrt{\eta[k]}}{\sqrt{b_{t-1}^2[k] + (\nabla_k f(w_t))^2 + \sigma^2}} - \nu_t[k]\right)\nabla_k f(w_t)g_t[k]\right] \leq \frac{\beta\sqrt{\eta[k]}\,(\nabla_k f(w_t))^2}{2\sqrt{b_{t-1}^2[k] + (\nabla f(w_t))^2 + \sigma^2}}
$$

$$
+ 2\beta\sqrt{\eta[k]}\sigma\mathbb{E}_t\left[\frac{g_t^2[k]}{b_t^2[k]}\right] \quad (25)
$$

*Proof.* Note that, using $\nu_t[k] \geq \frac{\beta\sqrt{\eta[k]}}{b_t[k]}$ we have

$$
\frac{\beta\sqrt{\eta[k]}}{\sqrt{b_{t-1}^2[k] + (\nabla_k f(w_t))^2 + \sigma^2}} - \nu_t[k]
$$

$$
\leq \beta\sqrt{\eta[k]}\left(\frac{1}{\sqrt{b_{t-1}^2[k] + (\nabla_k f(w_t))^2 + \sigma^2}} - \frac{1}{b_t[k]}\right)
$$

$$
= \beta\sqrt{\eta[k]}\left(\frac{b_t^2[k] - b_{t-1}^2[k] - (\nabla_k f(w_t))^2 - \sigma^2}{b_t[k]\sqrt{b_{t-1}^2[k] + (\nabla_k f(w_t))^2 + \sigma^2}\left(b_t[k] + \sqrt{b_{t-1}^2[k] + (\nabla_k f(w_t))^2 + \sigma^2}\right)}\right)
$$

$$
= \beta\sqrt{\eta[k]}\left(\frac{g_t^2[k] - (\nabla_k f(w_t))^2 - \sigma^2}{b_t[k]\sqrt{b_{t-1}^2[k] + (\nabla_k f(w_t))^2 + \sigma^2}\left(b_t[k] + \sqrt{b_{t-1}^2[k] + (\nabla_k f(w_t))^2 + \sigma^2}\right)}\right)
$$

$$
= \beta\sqrt{\eta[k]}\left(\frac{(g_t[k] + \nabla_k f(w_t))(g_t[k] - \nabla_k f(w_t)) - \sigma^2}{b_t[k]\sqrt{b_{t-1}^2[k] + (\nabla_k f(w_t))^2 + \sigma^2}\left(b_t[k] + \sqrt{b_{t-1}^2[k] + (\nabla_k f(w_t))^2 + \sigma^2}\right)}\right)
$$

$$
\leq \frac{\beta\sqrt{\eta[k]}\,|(g_t[k] + \nabla_k f(w_t))(g_t[k] - \nabla_k f(w_t))|}{b_t[k]\sqrt{b_{t-1}^2[k] + (\nabla_k f(w_t))^2 + \sigma^2}\left(b_t[k] + \sqrt{b_{t-1}^2[k] + (\nabla_k f(w_t))^2 + \sigma^2}\right)}
$$

$$
+ \frac{\beta\sqrt{\eta[k]}\sigma^2}{b_t[k]\sqrt{b_{t-1}^2[k] + (\nabla_k f(w_t))^2 + \sigma^2}\left(b_t[k] + \sqrt{b_{t-1}^2[k] + (\nabla_k f(w_t))^2 + \sigma^2}\right)}
$$

$$
\leq \frac{\beta\sqrt{\eta[k]}\,|g_t[k] - \nabla_k f(w_t)|}{b_t[k]\sqrt{b_{t-1}^2[k] + (\nabla_k f(w_t))^2 + \sigma^2}} + \frac{\beta\sqrt{\eta[k]}\sigma}{b_t[k]\sqrt{b_{t-1}^2[k] + (\nabla_k f(w_t))^2 + \sigma^2}}. \quad (26)
$$

Note that the second last inequality follows from the use of triangle inequality in the following way

$$
(g_t[k] + \nabla_k f(w_t))(g_t[k] - \nabla_k f(w_t)) - \sigma^2 \leq \left|(g_t[k] + \nabla_k f(w_t))(g_t[k] - \nabla_k f(w_t)) - \sigma^2\right|
$$

$$
\leq \left|(g_t[k] + \nabla_k f(w_t))(g_t[k] - \nabla_k f(w_t))\right| + \sigma^2,
$$

while the last inequality follows from

$$
b_t[k] + \sqrt{b_{t-1}^2[k] + (\nabla_k f(w_t))^2 + \sigma^2} \geq |g_t[k]| + |\nabla_k f(w_t)| \geq |g_t[k] + \nabla_k f(w_t)|,
$$

$$
b_t[k] + \sqrt{b_{t-1}^2[k] + (\nabla_k f(w_t))^2 + \sigma^2} \geq \sigma.
$$

Then from (26) we have

$$\mathbb{E}_t \left[ \left( \frac{\beta \sqrt{\eta[k]}}{\sqrt{b_{t-1}^2[k] + (\nabla_k f(w_t))^2 + \sigma^2}} - \nu_t[k] \right) \nabla_k f(w_t) g_t[k] \right]$$

$$\leq \quad \beta \sqrt{\eta[k]} \mathbb{E}_t \underbrace{\left[ \frac{|g_t[k] - \nabla_k f(w_t)| \, |\nabla_k f(w_t)| \, |g_t[k]|}{b_t[k] \sqrt{b_{t-1}^2[k] + (\nabla_k f(w_t))^2 + \sigma^2}} \right]}_{\text{term I}}$$

$$+ \beta \sqrt{\eta[k]} \mathbb{E}_t \underbrace{\left[ \frac{\sigma \, |\nabla_k f(w_t)| \, |g_t[k]|}{b_t[k] \sqrt{b_{t-1}^2[k] + (\nabla_k f(w_t))^2 + \sigma^2}} \right]}_{\text{term II}}. \quad (27)$$

For term I in (27), we use Lemma A.1 with

$$\lambda \quad = \quad \frac{2\sigma^2}{\sqrt{b_{t-1}^2[k] + (\nabla_k f(w_t))^2 + \sigma^2}},$$

$$a \quad = \quad \frac{|g_t[k]|}{b_t[k]},$$

$$b \quad = \quad \frac{|g_t[k] - \nabla_k f(w_t)| \, |\nabla_k f(w_t)|}{\sqrt{b_{t-1}^2[k] + (\nabla_k f(w_t))^2 + \sigma^2}},$$

to get

$$\beta \sqrt{\eta[k]} \mathbb{E}_t \left[ \frac{|g_t[k] - \nabla_k f(w_t)| \, |\nabla_k f(w_t)| \, |g_t[k]|}{b_t[k] \sqrt{b_{t-1}^2[k] + (\nabla_k f(w_t))^2 + \sigma^2}} \right]$$

$$\leq \quad \frac{\beta \sqrt{\eta[k]} \sqrt{b_{t-1}^2[k] + (\nabla_k f(w_t))^2 + \sigma^2}}{4\sigma^2} \frac{(\nabla_k f(w_t))^2 \, \mathbb{E}_t \left[ g_t[k] - \nabla_k f(w_t) \right]^2}{b_{t-1}^2[k] + (\nabla_k f(w_t))^2 + \sigma^2}$$

$$+ \frac{\beta \sqrt{\eta[k]} \sigma^2}{\sqrt{b_{t-1}^2[k] + (\nabla_k f(w_t))^2 + \sigma^2}} \mathbb{E}_t \left[ \frac{g_t^2[k]}{b_t^2[k]} \right]$$

$$\leq \quad \frac{\beta \sqrt{\eta[k]} (\nabla_k f(w_t))^2}{4\sqrt{b_{t-1}^2[k] + (\nabla_k f(w_t))^2 + \sigma^2}} + \beta \sqrt{\eta[k]} \sigma \mathbb{E}_t \left[ \frac{g_t^2[k]}{b_t^2[k]} \right]. \quad (28)$$

The last inequality follows from BV. Similarly, we again use Lemma A.1 with

$$\lambda \quad = \quad \frac{2}{\sqrt{b_{t-1}^2[k] + (\nabla_k f(w_t))^2 + \sigma^2}},$$

$$a \quad = \quad \frac{\sigma \, |g_t[k]|}{b_t[k]},$$

$$b \quad = \quad \frac{|\nabla_k f(w_t)|}{\sqrt{b_t^2[k] + (\nabla_k f(w_t))^2 + \sigma^2}}$$

and $\sqrt{b_t^2[k] + (\nabla_k f(w_t))^2 + \sigma^2} \geq \sigma$ to get

$$\beta \sqrt{\eta[k]} \mathbb{E}_t \left[ \frac{\sigma \, |\nabla_k f(w_t)| \, |g_t[k]|}{b_t[k] \sqrt{b_{t-1}^2[k] + (\nabla_k f(w_t))^2 + \sigma^2}} \right] \quad \leq \quad \beta \sqrt{\eta[k]} \sigma \mathbb{E}_t \left[ \frac{g_t^2[k]}{b_t^2[k]} \right]$$

$$+ \frac{\beta \sqrt{\eta[k]} (\nabla_k f(w_t))^2}{4\sqrt{b_{t-1}^2[k] + (\nabla f(w_t))^2 + \sigma^2}}. \quad (29)$$

Therefore using (28) and (29) in (28) we get

$$\mathbb{E}_t\left[\left(\frac{\beta\sqrt{\eta[k]}}{\sqrt{b_{t-1}^2[k] + (\nabla_k f(w_t))^2 + \sigma^2}} - \nu_t[k]\right)\nabla_k f(w_t)g_t[k]\right] \leq 2\beta\sqrt{\eta[k]}\sigma\mathbb{E}_t\left[\frac{g_t^2[k]}{b_t^2[k]}\right]$$

$$+ \frac{\beta\sqrt{\eta[k]}\,(\nabla_k f(w_t))^2}{2\sqrt{b_{t-1}^2[k] + (\nabla f(w_t))^2 + \sigma^2}}.$$

This completes the proof of this Lemma. $\qquad\square$

**Lemma A.7.**

$$\sum_{t=0}^{T}\frac{g_t^2[k]}{b_t^2[k]} \leq \log\left(\frac{b_T^2[k]}{b_0^2[k]}\right) + 1 \tag{30}$$

*Proof.* Using $b_t^2[k] = \sum_{\tau=0}^{t} g_\tau^2[k]$ we have

$$\sum_{t=0}^{T}\frac{g_t^2[k]}{b_t^2[k]} = 1 + \sum_{t=1}^{T}\frac{g_t^2[k]}{b_t^2[k]}$$

$$= 1 + \sum_{t=1}^{T}\frac{b_t^2[k] - b_{t-1}^2[k]}{b_t^2[k]}$$

$$= 1 + \sum_{t=1}^{T}\frac{1}{b_t^2[k]}\int_{b_{t-1}^2[k]}^{b_t^2[k]} dz$$

$$\leq 1 + \sum_{t=1}^{T}\int_{b_{t-1}^2[k]}^{b_t^2[k]}\frac{dz}{z}$$

$$= 1 + \int_{b_0^2[k]}^{b_T^2[k]}\frac{dz}{z}$$

$$= 1 + \log\left(\frac{b_T^2[k]}{b_0^2[k]}\right).$$

The inequality follows from the fact $\frac{1}{b_t^2[k]} \leq \frac{1}{z}$ when $b_{t-1}^2[k] \leq z \leq b_t^2[k]$. This completes the proof of the Lemma. $\qquad\square$

# B  Proof of Main Results

## B.1  Proof of Proposition 2.1

**Proposition B.1** (Scale invariance). Suppose we solve problems (4) and (5) using algorithm (6). Then, the iterates $\hat{w}_t$ and $\hat{w}_t^V$ corresponding to (4) and (5) follow: $\forall k \in [d]$

$$\hat{w}_{t+1}[k] = \hat{w}_t[k] - \frac{\beta m_t[k]}{\sum_{\tau=0}^t g_\tau^2[k]} g_t[k], \tag{31}$$

$$\hat{w}_{t+1}^V[k] = \hat{w}_t^V[k] - \frac{\beta m_t[k]}{\sum_{\tau=0}^t \left(g_\tau^V[k]\right)^2} g_t^V[k] \tag{32}$$

with $g_\tau = \varphi'_{i_\tau}(x_{i_\tau}^\top \hat{w}_\tau) x_{i_\tau}$ and $g_\tau^V = \varphi'_{i_\tau}(x_{i_\tau}^\top V \hat{w}_\tau) V x_{i_\tau}$ for $i_\tau$ chosen uniformly from $[n]$, $\tau = 0, 1, \ldots, t$, $t \geq 0$. Moreover, updates (31) and (32) satisfy

$$\hat{w}_t = V \hat{w}_t^V, \quad V g_t = g_t^V, \quad f(\hat{w}_t) = f^V(\hat{w}_t^V)$$

for all $t \geq 0$ when $\hat{w}_0 = \hat{w}_0^V = 0 \in \mathbb{R}^d$. Furthermore we have

$$\left\|g_t^V\right\|_{V^{-2}}^2 = \|g_t\|^2. \tag{33}$$

*Proof.* First, we will show $\hat{w}_t = V \hat{w}_t^V$ and $V g_t = g_t^V$ using induction. Note that for $\tau = 1$ and $k \in [d]$, we get

$$\hat{w}_1[k] = \frac{-\beta m_0[k] \varphi'_{i_0}(0) x_{i_0}[k]}{\left(\varphi'_{i_0}(0) x_{i_0}[k]\right)^2} = \frac{-\beta m_0[k]}{\varphi'_{i_0}(0) x_{i_0}[k]},$$

$$\hat{w}_1^V[k] = \frac{-\beta m_0[k] \varphi'_{i_0}(0) V_{kk} x_{i_0}[k]}{\left(\varphi'_{i_0}(0) V_{kk} x_{i_0}[k]\right)^2} = \frac{-\beta m_0[k]}{\varphi'_{i_0}(0) V_{kk} x_{i_0}[k]}.$$

as $\hat{w}_0 = \hat{w}_0^V = 0$. Therefore, we have $\forall k \in [d], \hat{w}_1[k] = V_{kk} \hat{w}_1^V[k]$. This can be equivalently written as $\hat{w}_1 = V \hat{w}_1^V$, as $V$ is a diagonal matrix. Then it is easy to check

$$V g_1 = \varphi'_{i_1}\left(x_{i_1}^\top \hat{w}_1\right) V x_{i_1} = \varphi'_{i_1}\left(x_{i_1}^\top V \hat{w}_1^V\right) V x_{i_1} = g_1^V, \tag{34}$$

where the second equality follows from $\hat{w}_1 = V \hat{w}_1^V$. Now, we assume the proposition holds for $\tau = 1, \cdots, t$. Then, we need to prove this proposition for $\tau = t + 1$. Note that, from (7) we have

$$\hat{w}_{t+1}[k] = \hat{w}_t[k] - \frac{\beta m_t[k]}{\sum_{\tau=0}^t g_\tau^2[k]} g_t[k] = V_{kk} \hat{w}_t^V[k] - \frac{\beta m_t[k] V_{kk}^2}{\sum_{\tau=0}^t (g_\tau^V[k])^2} \frac{g_t^V[k]}{V_{kk}} = V_{kk} \hat{w}_{t+1}^V[k].$$

Here, the second last equality follows from $\hat{w}_\tau = V \hat{w}_\tau^V$ and $V g_\tau = g_\tau^V \quad \forall \tau \in [t]$, while the last equality holds due to (32). Therefore, we have $\hat{w}_{t+1} = V \hat{w}_{t+1}^V$. Then similar to (34) we get $V g_{t+1} = g_{t+1}^V$ using $\hat{w}_{t+1} = V \hat{w}_{t+1}^V$. Again, using $\hat{w}_t = V \hat{w}_t^V$, we can rewrite $f(\hat{w}_t)$ as follow

$$f(\hat{w}_t) = \frac{1}{n} \sum_{i=1}^n \varphi_i\left(x_i^\top \hat{w}_t\right) = \frac{1}{n} \sum_{i=1}^n \varphi_i\left(x_i^\top V \hat{w}_t^V\right) = f^V(\hat{w}_t^V).$$

The last equality follows from (5). This proves $f(\hat{w}_t) = f^V(\hat{w}_t^V)$. Finally using $V g_t = g_t^V$ we get

$$\left\|g_t^V\right\|_{V^{-2}}^2 = \left(g_t^V\right)^\top V^{-2} g_t^V = g_t^\top V V^{-2} V g_t = \|g_t\|^2.$$

This completes the proof of Proposition 2.1. □

## B.2 Proof of Lemma 2.2

**Lemma B.2** (Decreasing step size). For $\nu_t[k]$ defined in (11) we have

$$\nu_{t+1}[k] \leq \nu_t[k] \qquad \forall k \in [d].$$

*Proof.* We want to show that $\nu_{t+1}[k] \leq \nu_t[k]$. Taking square and rearranging the terms (13) is equivalent to proving

$$b_t^4[k]m_{t+1}^2[k] \leq b_{t+1}^4[k]m_t^2[k]. \tag{35}$$

Using the expansion of $m_{t+1}^2[k], b_{t+1}^2[k]$, LHS of (35) can be expanded as follow

$$b_t^4[k]m_{t+1}^2[k] \quad = \quad b_t^4[k]\left(m_t^2[k] + \eta[k]g_{t+1}^2[k] + \frac{g_{t+1}^2[k]}{b_t^2[k] + g_{t+1}^2[k]}\right). \tag{36}$$

Similarly, the RHS of (35) can be expanded to

$$
\begin{aligned}
b_{t+1}^4[k]m_t^2[k] \quad &= \quad m_t^2[k]\left(b_t^2[k] + g_{t+1}^2[k]\right)^2 \\
&= \quad m_t^2[k]b_t^4[k] + m_t^2[k]g_{t+1}^4[k] + 2m_t^2[k]g_{t+1}^2[k]b_t^2[k].
\end{aligned}
\tag{37}
$$

Therefore using (36) and (37), inequality (35) is equivalent to

$$
\begin{aligned}
b_t^4[k]\left(m_t^2[k] + \eta[k]g_{t+1}^2[k] + \frac{g_{t+1}^2[k]}{b_t^2[k] + g_{t+1}^2[k]}\right) \quad &\leq \quad m_t^2[k]b_t^4[k] + m_t^2[k]g_{t+1}^4[k] \\
&\quad + 2m_t^2[k]g_{t+1}^2[k]b_t^2[k].
\end{aligned}
\tag{38}
$$

Now subtracting $m_t^2[k]b_t^4[k]$ from both sides of (38) and then multiplying both sides by $b_t^2[k] + g_{t+1}^2[k]$, (38) is equivalent to

$$
\begin{aligned}
\eta[k]g_{t+1}^2[k]b_t^6[k] + \eta[k]g_{t+1}^4[k]b_t^4[k] + g_{t+1}^2[k]b_t^4[k] \quad &\leq \quad m_t^2[k]g_{t+1}^4[k]b_t^2[k] + 2m_t^2[k]g_{t+1}^2[k]b_t^4[k] \\
&\quad + m_t^2[k]g_{t+1}^6[k] + 2m_t^2[k]g_{t+1}^4[k]b_t^2[k].
\end{aligned}
\tag{39}
$$

Therefore, proving (13) is equivalent to proving (39). Note that, from the expansion $m_t^2[k] = \eta[k]b_t^2[k] + \sum_{\tau=0}^{t}\frac{g_t^2[k]}{b_t^2[k]}$, we have $m_t^2[k] \geq \frac{g_0^2[k]}{b_0^2[k]} = 1$ and $m_t^2[k] \geq \eta[k]b_t^2[k]$. Then using $m_t^2[k] \geq 1$ we get

$$g_{t+1}^4[k]b_t^2[k] \leq m_t^2[k]g_{t+1}^4[k]b_t^2[k]. \tag{40}$$

Again, using $m_t^2[k] \geq \eta[k]b_t^2[k]$, we have

$$\eta[k]g_{t+1}^2[k]b_t^6[k] + \eta[k]g_{t+1}^4[k]b_t^4[k] \quad \leq \quad m_t^2[k]g_{t+1}^2[k]b_t^4[k] + m_t^2[k]g_{t+1}^4[k]b_t^2[k]. \tag{41}$$

Then adding (40) and (41) we get

$$\eta[k]g_{t+1}^2[k]b_t^6[k] + \eta[k]g_{t+1}^4[k]b_t^4[k] + g_{t+1}^2[k]b_t^4[k] \quad \leq \quad m_t^2[k]g_{t+1}^4[k]b_t^2[k] + 2m_t^2[k]g_{t+1}^2[k]b_t^4[k]. \tag{42}$$

Therefore, (39) is true due to (42) and $m_t^2[k]g_{t+1}^6[k] + 2m_t^2[k]g_{t+1}^4[k]b_t^2[k] \geq 0$. This completes the proof of the Lemma. $\square$

## B.3 Proof of Theorem 3.3

**Theorem B.3.** Suppose $f$ is $L$-smooth, $g_t = \nabla f(w_t)$ and $\eta, \beta$ are chosen such that $\nu_0[k] \leq \frac{1}{L}$ for all $k \in [d]$. Then for (11) we have

$$\min_{t \leq T} \|\nabla f(w_t)\|^2 \leq \frac{1}{T+1} \left( \sum_{k=1}^d b_0[k] + \frac{2(f(w_0) - f_*)}{\sqrt{\eta}\beta} \right)^2.$$

*Proof.* Suppose $g_t = \nabla f(w_t)$. Then using the smoothness of $f$ we get

$$
\begin{aligned}
f(w_{T+1}) &\leq f(w_T) + \langle g_T, w_{T+1} - w_T \rangle + \frac{L}{2} \|w_{T+1} - w_T\|^2 \\
&= f(w_T) + \sum_{k=1}^d g_T[k] (w_{T+1}[k] - w_T[k]) + \frac{L}{2} \sum_{k=1}^d (w_{T+1}[k] - w_T[k])^2 \\
&= f(w_T) - \sum_{k=1}^d \nu_T[k] g_T^2[k] + \frac{L}{2} \sum_{k=1}^d \nu_T^2[k] g_T^2[k] \\
&= f(w_T) - \sum_{k=1}^d \nu_T[k] \left( 1 - \nu_T[k] \frac{L}{2} \right) g_T^2[k].
\end{aligned}
$$

Then using this bound recursively we get

$$f(w_{T+1}) \leq f(w_0) - \sum_{t=0}^T \sum_{k=1}^d \nu_t[k] \left( 1 - \nu_t[k] \frac{L}{2} \right) g_t^2[k].$$

Note that, we initialized KATE such that $\nu_0[k] \leq \frac{1}{L} \forall k \in [d]$. Therefore using Lemma 2.2 we have $\nu_t[k] \leq \frac{1}{L}$, which is equivalent to $1 - \nu_t[k] \frac{L}{2} \geq \frac{1}{2}$ for all $k \in [d]$. Hence from (43) we have

$$f(w_{T+1}) \leq f(w_0) - \sum_{t=0}^T \sum_{k=1}^d \frac{\nu_t[k]}{2} g_t^2[k].$$

Then rearranging the terms and using $f(w_{T+1}) \geq f_*$ we get

$$\sum_{t=0}^T \sum_{k=1}^d \frac{\nu_t[k]}{2} g_t^2[k] \leq f(w_0) - f_*. \tag{43}$$

Then from (43) and $m_t[k] \geq \sqrt{\eta_0} b_t[k]$ we get

$$\sum_{t=0}^T \sum_{k=1}^d \frac{g_t^2[k]}{b_t[k]} \leq \frac{2(f(w_0) - f_*)}{\sqrt{\eta_0}\beta}. \tag{44}$$

Now from the definition of $b_t^2[k]$, we have $b_t^2[k] = b_{t-1}^2[k] + g_t^2[k]$. This can be rearranged to get

$$
\begin{aligned}
b_T[k] &= b_{T-1}[k] + \frac{g_T^2[k]}{b_T[k] + b_{T-1}[k]} \\
&\leq b_{T-1}[k] + \frac{g_T^2[k]}{b_T[k]} \tag{45} \\
&\leq b_0[k] + \sum_{t=0}^T \frac{g_t^2[k]}{b_t[k]}. \tag{46}
\end{aligned}
$$

Here the last inequality (46) follows from recursive use of (45). Then, taking squares on both sides and summing over $k \in [d]$ we get

$$
\begin{aligned}
\sum_{k=1}^d b_T^2[k] &\leq \sum_{k=1}^d \left( b_0[k] + \sum_{t=0}^T \frac{g_t^2[k]}{b_t[k]} \right)^2 \\
&\leq \left( \sum_{k=1}^d b_0[k] + \sum_{t=0}^T \sum_{k=1}^d \frac{g_t^2[k]}{b_t[k]} \right)^2 \\
&\leq \left( \sum_{k=1}^d b_0[k] + \frac{2(f(w_0) - f_*)}{\sqrt{\eta_0}\beta} \right)^2. \tag{47}
\end{aligned}
$$

The second inequality follows from $b_0[k] + \sum_{t=0}^{T} \frac{g_t^2[k]}{b_t[k]} \geq 0$ for all $k \in [d]$ and the last inequality from (44). Now note that $\sum_{t=0}^{T} \|g_t\|^2 = \sum_{t=0}^{T} \sum_{k=1}^{d} g_t^2[k] = \sum_{k=1}^{d} b_t^2[k]$. Therefore dividing both sides of (47) by $T + 1$, we get

$$\min_{t \leq T} \|\nabla f(w_t)\|^2 \leq \frac{1}{T+1} \left( \sum_{k=1}^{d} b_0[k] + \frac{2(f(w_0) - f_*)}{\sqrt{\eta_0}\beta} \right)^2.$$

This completes the proof of the theorem. □

## B.4  Proof of Theorem 3.4

**Theorem B.4.** Suppose $f$ is a $L$-smooth function and $g_t$ is an unbiased estimator of $\nabla f(w_t)$ such that BV holds. Moreover, we assume $\|\nabla f(w_t)\|^2 \leq \gamma^2$ for all $t$. Then KATE satisfies

$$\min_{t \leq T} \mathbb{E}\|\nabla f(w_t)\| \quad \leq \quad \left( \frac{\|g_0\|}{T} + \frac{2(\gamma + \sigma)}{\sqrt{T}} \right)^{1/2} \sqrt{\frac{2\mathcal{C}_f}{\beta \sqrt{\eta_0}}}$$

where

$$\mathcal{C}_f = f(w_0) - f_* + \sum_{k=1}^{d} \left( 2\beta \sqrt{\eta[k]}\sigma + \frac{\beta^2 \eta[k] L}{2} + \frac{\beta^2 L}{2 g_0^2[k]} \right) \left( \log \left( \frac{(\sigma^2 + \gamma^2)T}{g_0^2[k]} \right) + 1 \right).$$

*Proof.* Using smoothness, we have

$$
\begin{aligned}
f(w_{t+1}) \quad &\leq \quad f(w_t) + \langle \nabla f(w_t), w_{t+1} - w_t \rangle + \frac{L}{2} \|w_{t+1} - w_t\|^2 \\
&= \quad f(w_t) + \sum_{k=1}^{d} \nabla_k f(w_t) \left( w_{t+1}[k] - w_t[k] \right) + \frac{L}{2} \sum_{k=1}^{d} \left( w_{t+1}[k] - w_t[k] \right)^2 \\
&= \quad f(w_t) - \sum_{k=1}^{d} \nu_t[k] \nabla_k f(w_t) g_t[k] + \frac{L}{2} \sum_{k=1}^{d} \nu_t^2[k] g_t^2[k].
\end{aligned}
$$

Then, taking the expectation conditioned on $w_t$, we have

$$
\begin{aligned}
\mathbb{E}_t \left[ f(w_{t+1}) \right] \quad &\leq \quad f(w_t) - \sum_{k=1}^{d} \mathbb{E}_t \left[ \nu_t[k] \nabla_k f(w_t) g_t[k] \right] + \frac{L}{2} \sum_{k=1}^{d} \mathbb{E}_t \left[ \nu_t^2[k] g_t^2[k] \right] \\
&= \quad f(w_t) - \sum_{k=1}^{d} \mathbb{E}_t \left[ \nu_t[k] \nabla_k f(w_t) g_t[k] \right] + \frac{L}{2} \sum_{k=1}^{d} \mathbb{E}_t \left[ \nu_t^2[k] g_t^2[k] \right] \\
&\quad - \sum_{k=1}^{d} \frac{\beta \sqrt{\eta[k]}}{\sqrt{b_{t-1}^2[k] + (\nabla_k f(w_t))^2 + \sigma^2}} \mathbb{E}_t \left[ \nabla_k f(w_t) \left( \nabla_k f(w_t) - g_t[k] \right) \right] \\
&= \quad f(w_t) + \sum_{k=1}^{d} \mathbb{E}_t \left[ \left( \frac{\beta \sqrt{\eta[k]}}{\sqrt{b_{t-1}^2[k] + (\nabla_k f(w_t))^2 + \sigma^2}} - \nu_t[k] \right) \nabla_k f(w_t) g_t[k] \right] \\
&\quad + \frac{L}{2} \sum_{k=1}^{d} \mathbb{E}_t \left[ \nu_t^2[k] g_t^2[k] \right] - \sum_{k=1}^{d} \frac{\beta \sqrt{\eta[k]} (\nabla_k f(w_t))^2}{\sqrt{b_{t-1}^2[k] + (\nabla_k f(w_t))^2 + \sigma^2}}.
\end{aligned}
$$

The second last equality follows from $\mathbb{E}_t \left[ \nabla_k f(w_t) \left( \nabla_k f(w_t) - g_t[k] \right) \right] = \nabla_k f(w_t) \left( \nabla_k f(w_t) - \mathbb{E}_t \left[ g_t[k] \right] \right) = 0$. Now we use (25) to get

$$
\begin{aligned}
\mathbb{E}_t \left[ f(w_{t+1}) \right] \quad &\leq \quad f(w_t) + \sum_{k=1}^{d} 2\beta \sqrt{\eta[k]} \sigma \mathbb{E}_t \left[ \frac{g_t^2[k]}{b_t^2[k]} \right] + \frac{L}{2} \sum_{k=1}^{d} \mathbb{E}_t \left[ \nu_t^2[k] g_t^2[k] \right] \\
&\quad - \sum_{k=1}^{d} \frac{\beta \sqrt{\eta[k]} (\nabla_k f(w_t))^2}{2\sqrt{b_{t-1}^2[k] + (\nabla_k f(w_t))^2 + \sigma^2}}.
\end{aligned}
$$

Then rearranging the terms we have

$$
\begin{aligned}
\sum_{k=1}^{d} \frac{\beta \sqrt{\eta[k]} (\nabla_k f(w_t))^2}{2\sqrt{b_{t-1}^2[k] + (\nabla_k f(w_t))^2 + \sigma^2}} \quad &\leq \quad f(w_t) - \mathbb{E}_t \left[ f(w_{t+1}) \right] + \sum_{k=1}^{d} 2\beta \sqrt{\eta[k]} \sigma \mathbb{E}_t \left[ \frac{g_t^2[k]}{b_t^2[k]} \right] \\
&\quad + \frac{L}{2} \sum_{k=1}^{d} \mathbb{E}_t \left[ \nu_t^2[k] g_t^2[k] \right].
\end{aligned}
$$

Now we take the total expectations to derive

$$
\sum_{k=1}^{d} \mathbb{E}\left[\frac{\beta\sqrt{\eta[k]}\left(\nabla_k f(w_t)\right)^2}{2\sqrt{b_{t-1}^2[k] + \left(\nabla_k f(w_t)\right)^2 + \sigma^2}}\right] \leq \mathbb{E}\left[f(w_t)\right] - \mathbb{E}\left[f(w_{t+1})\right] + \sum_{k=1}^{d} 2\beta\sqrt{\eta[k]}\sigma\mathbb{E}\left[\frac{g_t^2[k]}{b_t^2[k]}\right]
$$

$$
+ \frac{L}{2}\sum_{k=1}^{d}\mathbb{E}\left[\nu_t^2[k]g_t^2[k]\right].
$$

The above inequality holds for any $t$. Therefore summing up from $t = 0$ to $t = T$ and using $f(w_{T+1}) \geq f_*$ we get

$$
\sum_{t=0}^{T}\sum_{k=1}^{d} \mathbb{E}\left[\frac{\beta\sqrt{\eta[k]}\left(\nabla_k f(w_t)\right)^2}{2\sqrt{b_{t-1}^2[k] + \left(\nabla_k f(w_t)\right)^2 + \sigma^2}}\right] \leq f(w_0) - f_* + \sum_{t=0}^{T}\sum_{k=1}^{d} 2\beta\sqrt{\eta[k]}\sigma\mathbb{E}\left[\frac{g_t^2[k]}{b_t^2[k]}\right]
$$

$$
+ \frac{L}{2}\sum_{t=0}^{T}\sum_{k=1}^{d}\mathbb{E}\left[\nu_t^2[k]g_t^2[k]\right]. \tag{48}
$$

Note that, using the expansion of $\nu_t^2[k]$ we have

$$
\nu_t^2[k] = \frac{\beta^2\eta[k]b_t^2[k] + \beta^2\sum_{j=0}^{t}\frac{g_j^2[k]}{b_j^2[k]}}{b_t^4[k]}
$$

$$
= \frac{\beta^2\eta[k]}{b_t^2[k]} + \frac{\beta^2}{b_t^4[k]}\sum_{j=0}^{t}\frac{g_j^2[k]}{b_j^2[k]}
$$

$$
\leq \frac{\beta^2\eta[k]}{b_t^2[k]} + \frac{\beta^2}{b_t^4[k]b_0^2[k]}\sum_{j=0}^{t}g_j^2[k] \tag{49}
$$

$$
= \frac{\beta^2\eta[k]}{b_t^2[k]} + \frac{\beta^2}{b_t^2[k]g_0^2[k]}. \tag{50}
$$

Here (49) follows from $b_j^2[k] \geq b_0^2[k]$ and (50) from $b_t^2[k] = \sum_{j=0}^{t}g_j^2[k]$. Then using (50) in (48) we derive

$$
\sum_{t=0}^{T}\sum_{k=1}^{d}\mathbb{E}\left[\frac{\beta\sqrt{\eta[k]}\left(\nabla_k f(w_t)\right)^2}{2\sqrt{b_{t-1}^2[k] + \left(\nabla_k f(w_t)\right)^2 + \sigma^2}}\right] \leq f(w_0) - f_* + \sum_{t=0}^{T}\sum_{k=1}^{d}\left(2\beta\sqrt{\eta[k]}\sigma + \frac{\beta^2\eta[k]L}{2} + \frac{\beta^2 L}{2g_0^2[k]}\right)\mathbb{E}\left[\frac{g_t^2[k]}{b_t^2[k]}\right]
$$

$$
\leq f(w_0) - f_*
$$

$$
+ \sum_{k=1}^{d}\left(2\beta\sqrt{\eta[k]}\sigma + \frac{\beta^2\eta[k]L}{2} + \frac{\beta^2 L}{2g_0^2[k]}\right)\mathbb{E}\left[\log\left(\frac{b_T^2[k]}{b_0^2[k]}\right) + 1\right].
$$

Here the last inequality follows from (30). Now using Jensen's Inequality (21) with $\Psi(z) = \log(z)$ we have

$$
\sum_{t=0}^{T}\sum_{k=1}^{d}\mathbb{E}\left[\frac{\beta\sqrt{\eta[k]}\left(\nabla_k f(w_t)\right)^2}{2\sqrt{b_{t-1}^2[k] + \left(\nabla_k f(w_t)\right)^2 + \sigma^2}}\right] \leq f(w_0) - f_*
$$

$$
+ \sum_{k=1}^{d}\left(2\beta\sqrt{\eta[k]}\sigma + \frac{\beta^2\eta[k]L}{2} + \frac{\beta^2 L}{2g_0^2[k]}\right)\left(\log\left(\frac{\mathbb{E}\left[b_T^2[k]\right]}{b_0^2[k]}\right) + 1\right).
$$

Now note that $\mathbb{E}\left[b_T^2[k]\right] = \sum_{t=0}^{T}\mathbb{E}\left[g_t^2[k]\right] = \sum_{t=0}^{T}\mathbb{E}\left[g_t[k] - \nabla_k f(w_t)\right]^2 + \left(\nabla_k f(w_t)\right)^2 \leq (\sigma^2 + \gamma^2)T$. Therefore, we have the bound

$$
\sum_{t=0}^{T}\sum_{k=1}^{d}\mathbb{E}\left[\frac{\beta\sqrt{\eta[k]}\left(\nabla_k f(w_t)\right)^2}{2\sqrt{b_{t-1}^2[k] + \left(\nabla_k f(w_t)\right)^2 + \sigma^2}}\right] \leq f(w_0) - f_* + 2\beta\sigma\sum_{k=1}^{d}\sqrt{\eta[k]}\log\left(\frac{e(\sigma^2 + \gamma^2)T}{b_0^2[k]}\right)
$$

$$
+ \sum_{k=1}^{d}\left(\frac{\beta^2\eta[k]L}{2} + \frac{\beta^2 L}{2g_0^2[k]}\right)\log\left(\frac{e(\sigma^2 + \gamma^2)T}{b_0^2[k]}\right). \tag{51}
$$

Here the RHS is exactly $\mathcal{C}_f$. Using (22) we have

$$\sum_{k=1}^{d} \frac{(\nabla_k f(w_t))^2}{\sqrt{b_{t-1}^2[k] + (\nabla_k f(w_t))^2 + \sigma^2}} \quad \geq \quad \frac{\sum_{k=1}^{d}(\nabla_k f(w_t))^2}{\sqrt{\sum_{k=1}^{d} b_{t-1}^2[k] + (\nabla_k f(w_t))^2 + \sigma^2}}$$

$$= \quad \frac{\|\nabla f(w_t)\|^2}{\sqrt{\|b_{t-1}\|^2 + \|\nabla f(w_t)\|^2 + d\sigma^2}}. \tag{52}$$

Therefore using (52) in (51) we arrive at

$$\sum_{t=0}^{T} \mathbb{E}\left[\frac{\|\nabla f(w_t)\|^2}{\sqrt{\|b_{t-1}\|^2 + \|\nabla f(w_t)\|^2 + d\sigma^2}}\right] \quad \leq \quad \frac{2\mathcal{C}_f}{\beta\sqrt{\eta_0}}. \tag{53}$$

Now we use Holder's Inequality (20) $\frac{\mathbb{E}(XY)}{\left(\mathbb{E}|Y|^3\right)^{\frac{1}{3}}} \leq \left(\mathbb{E}|X|^{\frac{3}{2}}\right)^{\frac{2}{3}}$ with

$$X = \left(\frac{\|\nabla f(w_t)\|^2}{\sqrt{\|b_{t-1}\|^2 + \|\nabla f(w_t)\|^2 + d\sigma^2}}\right)^{\frac{2}{3}} \quad \text{and} \quad Y = \left(\sqrt{\|b_{t-1}\|^2 + \|\nabla f(w_t)\|^2 + d\sigma^2}\right)^{\frac{2}{3}}$$

to get a lower bound on LHS of (53):

$$\mathbb{E}\left[\frac{\|\nabla f(w_t)\|^2}{\sqrt{\|b_{t-1}\|^2 + \|\nabla f(w_t)\|^2 + d\sigma^2}}\right] \quad \geq \quad \frac{\mathbb{E}\left[\|\nabla f(w_t)\|^{\frac{4}{3}}\right]^{\frac{3}{2}}}{\sqrt{\mathbb{E}\left(\|b_{t-1}\|^2 + \|\nabla f(w_t)\|^2 + d\sigma^2\right)}}$$

$$\geq \quad \frac{\mathbb{E}\left[\|\nabla f(w_t)\|^{\frac{4}{3}}\right]^{\frac{3}{2}}}{\sqrt{\|b_0\|^2 + 2t(\gamma^2 + d\sigma^2)}}. \tag{54}$$

Therefore from (53) and (54) we get

$$\frac{T}{\sqrt{\|b_0\|^2 + 2T(\gamma^2 + d\sigma^2)}} \min_{t \leq T} \mathbb{E}\left[\|\nabla f(w_t)\|^{\frac{4}{3}}\right]^{\frac{3}{2}} \quad \leq \quad \frac{2\mathcal{C}_f}{\beta\sqrt{\eta_0}}.$$

Then multiplying both sides by $\frac{\|b_0\| + \sqrt{2T}(\gamma + \sqrt{d}\sigma)}{T}$ we have

$$\min_{t \leq T} \mathbb{E}\left[\|\nabla f(w_t)\|^{\frac{4}{3}}\right]^{\frac{3}{2}} \quad \leq \quad \left(\frac{\|b_0\|}{T} + \frac{2(\gamma + \sigma)}{\sqrt{T}}\right)\frac{2\mathcal{C}_f}{\beta\sqrt{\eta_0}}.$$

Here we use $\mathbb{E}\left[\|\nabla f(w_t)\|\right]^{\frac{4}{3}} \leq \mathbb{E}\left[\|\nabla f(w_t)\|^{\frac{4}{3}}\right]$ (follows from Jensen's Inequality (21) with $\Psi(z) = z^{4/3}$) in the above equation to get

$$\min_{t \leq T} \mathbb{E}\left[\|\nabla f(w_t)\|\right]^2 \quad \leq \quad \left(\frac{\|b_0\|}{T} + \frac{2(\gamma + \sigma)}{\sqrt{T}}\right)\frac{2\mathcal{C}_f}{\beta\sqrt{\eta_0}}.$$

This completes the proof of the Theorem. $\qquad\square$

# C   Additional Experiments: Scale-Invariance Verification

In this experiment, we implement KATE on problems (4) (for unscaled data) and (5) (for scaled data) with

$$\varphi_i(z) = \log\left(1 + \exp\left(-y_i z\right)\right).$$

We generate the data similar to Section 4.1.1. We run KATE for $10,000$ iterations with mini-batch size 10, $\eta = {}^1/(\nabla f(w_0))^2$ and plot functional value $f(w_t)$ and accuracy in Figures 11a and 11b. We use the proportion of correctly classified samples to compute accuracy, i.e. $\frac{1}{n}\sum_{i=1}^{n} \mathbf{1}_{\left\{y_i x_i^\top w_t \geq 0\right\}}$.

In plots 11a and 11b, the functional value and accuracy of KATE coincide, which aligns with our theoretical findings (Proposition 2.1). Figure 11c plots $\|\nabla f(w_t)\|^2$ and $\|\nabla f(w_t)\|_{V^{-2}}^2$ for unscaled and scaled data respectively. Here, (10) explains the identical values taken by the corresponding gradient norms of KATE iterates for the scaled and unscaled data. Similarly, in Figure 12, we compare the performance of AdaGrad on scaled and un-scaled data. This figure illustrates the lack of the scale-invariance for AdaGrad.

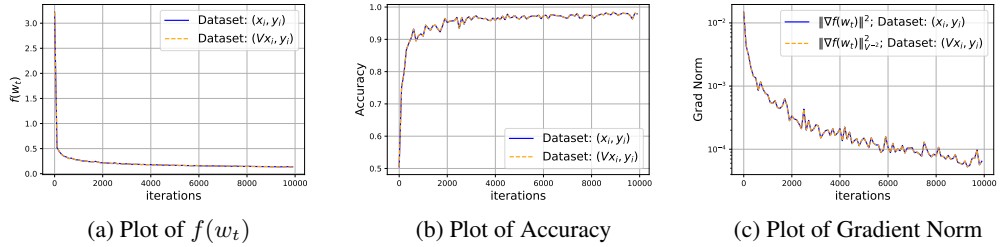

(a) Plot of $f(w_t)$      (b) Plot of Accuracy      (c) Plot of Gradient Norm

Figure 11: Comparison of KATE on scaled and un-scaled data. Figures 11a, and 11b plot the functional value $f(w_t)$ and accuracy on scaled and unscaled data, respectively. Figure 11c plots $\|\nabla f(w_t)\|^2$ and $\|\nabla f(w_t)\|_{V^{-2}}^2$ for unscaled and scaled data respectively.

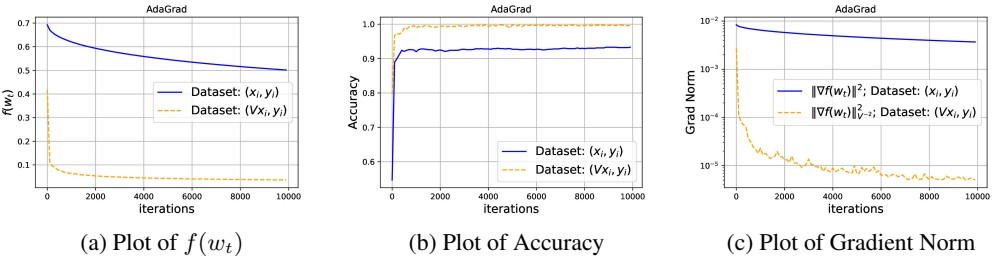

(a) Plot of $f(w_t)$      (b) Plot of Accuracy      (c) Plot of Gradient Norm

Figure 12: Comparison of AdaGrad on scaled and un-scaled data. Figures 12a, and 12b plot the functional value $f(w_t)$ and accuracy on scaled and unscaled data, respectively. Figure 12c plots $\|\nabla f(w_t)\|^2$ and $\|\nabla f(w_t)\|_{V^{-2}}^2$ for unscaled and scaled data respectively.

