# OpenReview forum: "Remove that Square Root: A New Efficient Scale-Invariant Version of AdaGrad"
_NeurIPS.cc/2024/Conference — NeurIPS 2024 poster_

### Official Review · Reviewer_uk2H · 2024-07-07

**Soundness:** 2
**Presentation:** 2
**Contribution:** 2
**Rating:** 4
**Confidence:** 2

**Summary:**

This paper presents a novel algorithm, KATE, which demonstrates impressive scale-invariance properties for Generalized Linear Models.

**Strengths:**

This paper presents a novel algorithm, KATE, which demonstrates impressive scale-invariance properties for Generalized Linear Models. The thorough theoretical analysis provided in the paper, along with the experimental results, showcases the effectiveness of KATE in various machine learning tasks. The comparison with existing algorithms like AdaGrad and Adam highlights the superior performance of KATE in terms of convergence rate and efficiency.

**Weaknesses:**

While the paper excels in presenting the theoretical foundations and empirical results of the KATE algorithm, there are a few areas that could be further strengthened. Firstly, the paper could benefit from a more detailed discussion on the limitations of the proposed algorithm, especially in scenarios where certain assumptions may not hold. Additionally, providing insights into the computational efficiency and scalability of KATE with larger datasets could enhance the practical applicability of the algorithm.

**Questions:**

1 How does the scale-invariance property of KATE impact its performance in real-world applications compared to traditional adaptive algorithms?

2 Can the authors elaborate on the computational complexity of KATE and how it scales with the size of the dataset?

3 Are there any specific scenarios or types of machine learning tasks where KATE may not perform as effectively, and if so, how does the algorithm address these limitations?

---

> ### Author Rebuttal · Authors · 2024-08-07
>
> We thank the reviewer for a detailed review. Below, we address the reviewer's questions and concerns.
>
> > **Firstly, the paper could benefit from a more detailed discussion of the limitations of the proposed algorithm, especially in scenarios where certain assumptions may not hold.**
>
> Thank you for your suggestion. We acknowledge that one of the assumptions in our work is the bounded gradient assumption, which states that $|| \nabla f(w_t) ||^2 \leq \gamma^2$. This assumption is strong and critical for our convergence analysis of the KATE algorithm in the stochastic setting, as presented in Theorem 3.4. This assumption may not hold in several scenarios when the optimization problem's domain is unbounded, and the gradient might grow without limit. To address this limitation, we plan to investigate using gradient clipping techniques with KATE in future work.
>
> However, we emphasize that this assumption is not a limitation of our work. The bounded gradient assumption is a standard regularity condition employed in analyzing numerous stochastic optimization algorithms, including well-established methods like Adam and AdaGrad. The bounded gradient assumption helps manage the stochastic noise and ensure the stability of the updates, which is crucial for deriving theoretical guarantees. By adopting this assumption, we align our analysis with the existing body of work, making our results comparable and sticking to established theoretical frameworks. We will add the above discussion in the updated version.
>
> > **Additionally, providing insights into the computational efficiency and scalability of KATE with larger datasets could enhance the practical applicability of the algorithm.**
> > **Can the authors elaborate on the computational complexity of KATE and how it scales with the size of the dataset?**
>
> We thank the reviewer for their insightful question regarding the computational efficiency and scalability of KATE. To address computational efficiency, KATE is designed to be computationally efficient by requiring only one gradient computation per iteration. This is achieved by storing and reusing intermediate computations, specifically $m^2_{t-1}$​ and $b^2_{t-1}$​, from the previous iteration. This approach avoids redundant calculations and contributes to the algorithm's overall efficiency. In terms of convergence rate, KATE exhibits a favorable theoretical convergence rate of $O(\log T/ \sqrt{T})$ in the stochastic setting and $O(1/T)$ in the deterministic setting. A detailed comparison of convergence rates with other algorithms is provided in Table 1 of our paper.
>
> To demonstrate KATE's scalability, we have conducted experiments on large-scale datasets, including CIFAR-10 for image classification and the emotions dataset from Hugging Face Hub for BERT fine-tuning. Our results indicate that KATE outperforms both Adam and AdaGrad on the CIFAR-10 dataset while exhibiting comparable performance to Adam on the emotions dataset. These findings suggest that KATE is capable of handling large-scale datasets efficiently and effectively. **We believe that the combination of computational efficiency, favorable convergence rate, and strong empirical performance on large-scale datasets highlights KATE's potential for practical applications.**
>
>
> >**How does the scale-invariance property of KATE impact its performance in real-world applications compared to traditional adaptive algorithms?**
>
> Thank you for the question. We believe that good adaptive methods should work well for all problems, particularly relatively simple problems that we can analyze. Therefore, we see scale-invariance as an intermediate step toward designing practical methods for large classes of ML problems, going beyond generalized linear models. As discussed in Section 2: Motivation and Algorithm Design, KATE was specifically designed to make AdaGrad scale-invariant. This design choice is pivotal, as it allows KATE to outperform AdaGrad across all tasks on real-world datasets, such as CIFAR-10 and the emotions dataset from Hugging Face Hub. We attribute KATE's superior performance to its scale-invariance property, which ensures consistent and reliable results regardless of the scale of the gradients.
>
> Additionally, we compared KATE with Adam in our experiments. While KATE does not incorporate momentum and the Exponential Moving Average (EMA) of second moments (also mentioned by reviewer eTbr), as Adam does, it still performs comparably to or better than Adam in real-world applications. This observation underscores the potential for further enhancements. If a version of Adam could be designed to be scale-invariant, similar to KATE, it might achieve even better performance.
>
> In summary, our work with KATE not only demonstrates its effectiveness but also lays the groundwork for future research in adaptive methods. By leveraging the scale-invariance property, there is significant potential for further advancements in this field.
>
>
> > **Are there any specific scenarios or types of machine learning tasks where KATE may not perform as effectively, and if so, how does the algorithm address these limitations?**
>
> Thank you for the question. Currently, we are not aware of any situations where KATE performs worse than AdaGrad or Adam. As we have highlighted earlier, we believe that KATE's scale-invariance property plays a crucial role in ensuring its superior performance.
>
> **If you agree that we addressed all issues, please consider raising your score. If you believe this is not the case, please let us know so that we can respond.**

---

### Official Review · Reviewer_bngL · 2024-07-11

**Soundness:** 3
**Presentation:** 4
**Contribution:** 3
**Rating:** 7
**Confidence:** 4

**Summary:**

This work proposes an optimizer that achieves the optimal convergence guarantee for smooth nonconvex settings and more importantly the scale-invariance property.

**Strengths:**

The paper is very cleanly written. It was very easy to follow.
The main results look sound, and the experimental results are well presented as well.

**Weaknesses:**

Overall, the paper presents the main scope and the results very well.
In other words, I do not have much concern for this paper by itself.

My only concern is the importance of the problem it tackles. Scale-invariance is definitely a desirable property, but I'm not sure what advances it brings about for the ML optimization. What I mean is, in general, the important question in the community is whether one can design an optimizer that has a noticeable advantage over the previous popular ones. I'm not sure whether resolving scale-invariance will drastically improve our current technology for optimizing ML models. The experiments presented in this paper doesn't seem to justify this in a compelling way. (I think results are convex models have limited practical impact.)

**Questions:**

- I think since a lot of baselines this paper considers is originally designed for online convex optimization, it is important to also do an extensive analysis of KATE for the corresponding OCO settings and compare. In particular, some regret analysis as well as the convergence analysis for nonsmooth convex setting might be helpful.


- The main motivation this paper argues for the need of scale-invariance under the data scaling is that previous approaches could be brittle when data has poor-scaling or ill-conditioning. In order to make a case that this is an important question to solve, I'd like to see some practically relevant scenarios where the lack of good scaling of data leads to failure of non-scale-invariance approaches like Adam and Adagrad. I think for this paper to be have a bigger impact, a compelling set of experiments along this line seems to be necessary.

- Do you expect poor data scaling to be the major issue in the training large AI models such as LLMs? If that's the case, I think this paper might have a bigger impact.

**Limitations:**

As I said, the reason for the current score is mainly due to the main scope of this paper.
In my opinion, unless the authors have compelling experimental results or arguments, tacking the scale-invarance seems to have limited practical impact in the community.

---

> ### Author Rebuttal · Authors · 2024-08-07
>
> We thank the reviewer for a detailed review. Below, we address the reviewer's questions and concerns.
>
> > **My only concern is the importance of the problem it tackles. Scale-invariance is definitely a desirable property, but I'm not sure what advances it brings about for ML optimization. What I mean is, in general, the important question in the community is whether one can design an optimizer that has a noticeable advantage over the previous popular ones. I'm not sure whether resolving scale invariance will drastically improve our current technology for optimizing ML models. The experiments presented in this paper don't seem to justify this in a compelling way. (I think results are convex models that have limited practical impact.)**
>
> Thank you for your insightful comment. We believe that good adaptive methods should work well for all problems, particularly relatively simple problems that we can analyze. Therefore, we see scale-invariance as an intermediate step toward designing practical methods for large classes of ML problems, going beyond generalized linear models. We fully acknowledge the reviewer's concern and would like to emphasize the significant performance improvements that KATE offers compared to the AdaGrad algorithm. Specifically, KATE has consistently outperformed AdaGrad across various tasks on real-world datasets, such as CIFAR-10 and the emotions dataset from Hugging Face Hub. We attribute KATE's superior performance to its scale-invariance property. Moreover, it is noteworthy that KATE performs comparably to Adam on these tasks, even though it does not incorporate advanced techniques like momentum and Exponential Moving Average (EMA). This observation suggests that if we were to develop a scale-invariant version of Adam, it could potentially outperform the standard Adam algorithm across all tasks.
>
> In summary, our paper lays the groundwork for the future development of improved scale-invariant algorithms. We strongly believe this work represents an important stride towards developing more robust and effective optimization methods.
>
> >**I think since a lot of baselines this paper considers are originally designed for online convex optimization, it is important to also do an extensive analysis of KATE for the corresponding OCO settings and compare. In particular, some regret analysis and convergence analysis for non-smooth convex settings might be helpful.**
>
> We thank the reviewer for the suggestion. We believe that our analysis can be generalized to the case of online convex optimization and non-smooth convex setting, and up to the logarithmic factor, we can derive the standard $1/\sqrt{T}$ rate for these settings. Therefore, in terms of the theoretical convergence bounds, KATE has comparable convergence rates with the best-known algorithms in these settings (up to the logarithmic factor). However, it would be interesting to compare KATE with other methods designed for online convex optimization in the experiments (and, perhaps, take the best of two worlds to develop even better methods). We leave this direction for future work.
>
> > **The main motivation this paper argues for the need for scale-invariance under the data scaling is that previous approaches could be brittle when data has poor scaling or ill-conditioning. In order to make a case that this is an important question to solve, I'd like to see some practically relevant scenarios where the lack of good scaling of data leads to the failure of non-scale-invariance approaches like Adam and Adagrad. I think for this paper to have a bigger impact, a compelling set of experiments along this line seems to be necessary.**
>
> It is common practice to normalize/scale the input data to get better results in Neural Network training (e.g., see Horváth and Richtárik (2020)). We are currently working on extending our numerical experiments for image classification to illustrate how the methods behave with and without data normalization and with and without scaling.
>
> >**Do you expect poor data scaling to be the major issue in training large AI models such as LLMs? If that's the case, I think this paper might have a bigger impact.**
>
> That is an interesting question. We have not yet considered this aspect in our current work. Additionally, our research is primarily theoretical, and conducting experiments on large language models (LLMs) falls outside the scope of this paper. However, we recognize the importance of the reviewer's concern and commit to exploring the existing literature on LLMs to investigate if data scaling can impact the training of large AI models. We agree that if such an impact is found, it could represent a significant contribution to the community.
>
> > **As I said, the reason for the current score is mainly due to the main scope of this paper. In my opinion, unless the authors have compelling experimental results or arguments, tacking the scale-invariance seems to have limited practical impact in the community.**
>
> Please see our response to the weaknesses.
>
> **Thanks for the valuable suggestions and the positive evaluation. If you agree that we addressed all issues, please consider raising your score to support our work. If you believe this is not the case, please let us know so we can respond.**

---

> > ### Comment · Reviewer_bngL · 2024-08-09
> > **Thanks**
> >
> > Thanks for your responses.
> > I read through your responses, and I understand that the scale invariance could be beneficial in practice.
> > I'll increase my score to 7.

---

### Official Review · Reviewer_eTbr · 2024-07-12

**Soundness:** 3
**Presentation:** 4
**Contribution:** 3
**Rating:** 6
**Confidence:** 4

**Summary:**

The paper introduces a novel optimization algorithm which demonstrate scale-invariance property for generlized linear models unlike Adagrad. The authors analyzed KATE for smooth and non-convex functions and on generalized linear models to obtain the same convergence upper bounds (asymptotically) as Adagrad and Adam. However the scale invariant algorithm implies, the speed of the algorithm would be the same as for the best scaling of data. The authors also empirically verify this via numerical experiments on logistic regression and deep learning tasks.

**Strengths:**

1) The authors come up with KATE which is scale invariant, and yet admit a convergence upper bound of $\mathcal{O}(\log(T)/\sqrt{T})$, which is an interesting algorithmic finding.
2) The paper is easy to read and understand.
3) Comprehensive appendix.

**Weaknesses:**

I think authors need to emphasize the first point in Strengths by comparing with the diagonal online-newton method (diag-SONew) [1] which is also scale-invariant algorithm and what part of the analysis can fail for diag-SONew. I wrote the algorithm without the EMA here, which is essentially adagrad without the square root but with varying $\eta_t$:

$$
w_t := w_{t-1} -\eta_t g_t /({\sum_{s=1}^t g_s* g_s})
$$

 schedules one can try is $\eta_t=1$ or $\eta_t=\sqrt{t}$. I think comparing with the latter schedule will give a good understanding of the novelty behind KATE, as the schedule simulates the square root in Adagrad while being scale-invariant. Similarly empirical comparisons (if possible) with this simple algorithm in logistic-regression can help understand which algorithm is better.

2) Comparison with Adam doesn’t make sense in neural networks as KATE lacks momentum and EMA in second-moments (which are key features of Adam). Devising KATE with these features similar to [2] would help the empirical performance in neural networks.

[1] Devvrit, Fnu, Sai Surya Duvvuri, Rohan Anil, Vineet Gupta, Cho-Jui Hsieh, and Inderjit Dhillon. "A computationally efficient sparsified online newton method." Advances in Neural Information Processing Systems 36 (2024).
[2] Defazio, A., & Mishchenko, K. (2023, July). Learning-rate-free learning by d-adaptation. In International Conference on Machine Learning (pp. 7449-7479). PMLR.

**Questions:**

In line 80-81  “meaning that the speed of convergence of KATE is the same as for the best scaling of the data.”, is this reflected in the convergence bound? i.e., whether scale dependence of Adagrad bound vs scale independence of KATE bound is emphasized. The Table 1 only mention asymptotic bounds for Adagrad. It would help the paper if the convergence bounds are analyzed for the generalized linear models for Adagrad and KATE to understand the affect of scale on the final upperbounds.

**Limitations:**

Authors have adequately addressed limitations.

---

> ### Author Rebuttal · Authors · 2024-08-07
>
> We thank the reviewer for a detailed review and positive evaluation. Below, we address the reviewer's questions and concerns.
>
> > **I think authors need to emphasize the first point in Strengths by comparing with the diagonal online-newton method (diag-SONew) [1], which is also a scale-invariant algorithm, and what part of the analysis can fail for diag-SONew….. schedules one can try is $\eta_t = 1$ or $\eta_t = \sqrt{t}$. I think comparing with the latter schedule will give a good understanding of the novelty behind KATE, as the schedule simulates the square root in Adagrad while being scale-invariant. Similarly, empirical comparisons (if possible) with this simple algorithm in logistic regression can help understand which algorithm is better.**
>
> Thank you for your excellent question. Indeed, the design of our KATE optimizer was motivated by a similar exploration of different step-size strategies. Our approach involved experimenting with different step sizes to determine their effectiveness before proving the corresponding theorems. Initially, we tried setting $\eta_t=1$, but it did not converge to the optimal solution (as noted on lines 123-124 of our paper).
>
> We then experimented with $\eta_t = \sqrt{t}$​, which showed mixed results—performing well in some experiments but poorly in others. From these experiments, we observed that $\eta_t = \eta \sqrt{t}$​, where $\eta$ needs to be tuned, performed better. However, this algorithm was still not robust to the choice of $\eta$, unlike KATE. In particular, when full gradients are used instead of stochastic ones, the denominator is provably bounded. This means that with $\eta_t = \eta \sqrt{t}$, the overall stepsize is growing, and one has to choose $\eta$ small enough (and dependent on the time horizon) to avoid the divergence. For KATE, the numerator of the step size is adaptive and depends on the function $f$ and stochastic gradients, allowing it to adjust to the problem more effectively.
>
> We will incorporate this discussion and include experiments comparing KATE with the step sizes you mentioned in the updated version of our paper. We hope this explanation clarifies our design process and KATE's robustness in handling different step sizes.
>
> >**Comparison with Adam doesn’t make sense in neural networks as KATE lacks momentum and EMA in the second moments (which are key features of Adam). Devising KATE with features similar to [2] would help with empirical performance in neural networks.**
>
> Thank you for this insightful comment. We are aware of the D-adaptation work and recognise its importance. We plan to incorporate momentum and Exponential Moving Average (EMA) into KATE as part of our future research. This enhancement is on our to-do list, and we are excited about the potential improvements it can bring to KATE's performance and robustness. Thank you for highlighting this area, and we look forward to sharing our progress in future publications.
>
> >**In lines 80-81, “meaning that the speed of convergence of KATE is the same as for the best scaling of the data.” is this reflected in the convergence bound? i.e., the scale dependence of the Adagrad bound vs. the scale independence of the KATE bound is emphasised. Table 1 only mentions asymptotic bounds for Adagrad. It would help the paper if the convergence bounds were analysed for the generalised linear models for Adagrad and KATE to understand the effect of scale on the final upper bounds.**
>
> We thank the reviewer for the great question. Since our convergence bounds are derived for the general smooth non-convex problems, which are not necessarily generalised linear models (GLMs), the scale-invariance is not reflected in the convergence bounds. Adapting our analysis to the case of GLMs is an interesting direction for future research.
>
> **Thanks for the valuable suggestions and the positive evaluation. If you agree that we addressed all issues, please consider raising your score to support our work. If you believe this is not the case, please let us know so we can respond.**

---

> > ### Comment · Reviewer_eTbr · 2024-08-13
> >
> > I thank the authors for their rebuttal, and I would like to maintain my score.

---

### Official Review · Reviewer_ZFDF · 2024-07-19

**Soundness:** 3
**Presentation:** 3
**Contribution:** 3
**Rating:** 7
**Confidence:** 4

**Summary:**

This paper proposes a scale-invariant variant of AdaGrad, called KATE, particularly for generalized linear models. Theoretically, the authors proved a convergence rate of $\mathcal{O}(\log T/\sqrt{T})$ for KATE, matching the best known rates for AdaGrad and Adam. Numerical experiments are used to illustrate KATE on several machine learning tasks, which outperforms AdaGrad consistently and matches/outperforms Adam.

**Strengths:**

This work studies a crucial problem of developing a scale-invariant variant of AdaGrad which is particularly useful whenever data exhibit poor scaling or ill-conditioning. Convergence rates of KATE similar to those of AdaGrad and Adam under both deterministic and stochastic settings are established, with comprehensive numerical experiments to justify the efficacy of KATE compared to AdaGrad and Adam.

**Weaknesses:**

As the motivation of KATE is to develop a scale-invariant optimizer, the experiments (even the one with simulated data) do not seem to have demonstrated this.

**Questions:**

While it could be harder to demonstrate the scale-invariant property of KATE with real data experiments, is it possible to demonstrate the scale invariant property of KATE compared to AdaGrad and Adam with the logistic regression using simulated data? This could better help readers understand why the scale-invariant property of KATE is plausible compared to other adaptive optimizers.

**Limitations:**

Yes.

---

> ### Author Rebuttal · Authors · 2024-08-07
>
> We thank the reviewer for a detailed review and positive evaluation. Below, we address the reviewer's questions and concerns.
>
> > **While it could be harder to demonstrate the scale-invariant property of KATE with real data experiments, is it possible to demonstrate the scale-invariant property of KATE compared to AdaGrad and Adam with the logistic regression using simulated data?
> This could better help readers understand why the scale-invariant property of KATE is plausible compared to other adaptive optimizers.**
>
> Thank you for your insightful feedback. We appreciate the reviewer's concern regarding the demonstration of KATE's scale-invariance property. To address this, we have included plots that illustrate the scale-invariance of KATE using simulated data. These plots can be found in Appendix C, Figure 11.
>
> These plots provide a visual demonstration of the scale-invariance property that we have theoretically proved in Proposition 2.1. Unfortunately, due to space constraints, we were unable to include these plots in the main body of the paper. However, we have referenced them in the main paper on line 121 to ensure that readers are aware of their existence and can easily locate them.
>
> We hope these additional visual proofs address your concern. If this clarification meets your expectations, please consider raising your score to further support our work.

---

> > ### Comment · Reviewer_ZFDF · 2024-08-12
> > **Response to Rebuttal**
> >
> > Thanks so much to the authors for their response.
> >
> > I think I did miss the pointers and did not go through the appendix during my review, so thanks for your pointer. While I acknowledge the significance of this work, I do think that experiments of larger scales such as ImageNet (and open-source softwares if possible) are required to justify a higher score. Therefore I've decided to maintain my rating.

---

### Author Rebuttal · Authors · 2024-08-07

We thank the reviewers for their feedback and time. We appreciate that the reviewers acknowledged the following strengths of our work:

- Reviewer ZFDF recognises the importance of developing a scale-invariant version of AdaGrad.
- Reviewer uk2H finds the scale-invariance property of KATE very impressive.
- Reviewer eTbr highlights the significance of KATE achieving the $O(\log T/ \sqrt{T})$ rate even with the scale-invariance property.
- Reviewers eTbr and bngL find our work easy to read and understand. They appreciate the paper's presentation.
- Reviewers eTbr, ZFDF, and uk2H value the thorough theoretical analysis of KATE provided in the paper.
- All the reviewers acknowledge the numerical experiments provided in our paper.

In our responses, we have addressed the reviewers' questions and concerns in detail. If the reviewers have further questions/concerns/comments, we will be happy to participate in the discussion.

---

### Comment · Area_Chair_kww2 · 2024-08-12
**Discussion with the authors**

Dear reviewers:

As the discussion period is going to end soon, please try to actively engage with the authors about the paper. Thanks a lot for your help and dedication.

You AC.

---

### Decision · Program_Chairs · 2024-09-25

**Decision:**

Accept (poster)

**Comment:**

The paper introduces KATE, a new optimization algorithm that adapts the AdaGrad method, for Generalized Linear Models. KATE is proven to have a convergence rate comparable to AdaGrad and Adam for smooth non-convex problems. Experimental results show that KATE consistently outperforms AdaGrad and matches or exceeds Adam's performance in various machine learning tasks, including image and text classification. Most of the reviewers recognized that this is an important work in providing a new adaptive stepsize rule that are scale invariant. However, some reviewers also pointed out, and I agree, that larger scale and more practical numerical experiments are needed to truly demonstrate the proposed approach. I would suggest that the authors include such numerical tests, if at all possible.